# Iran's great scientific divergence: The Middle-Income trap and the Political Economy of Science Policy

Ehsan Roohi *

Mechanical and Industrial Engineering, University of Massachusetts Amherst, Amherst, Massachusetts, United States of America

* roohie@umass.edu

## Abstract

This study examines how the 1979 Iranian Revolution altered Iran's long-term scientific development using publication and citation data for 1960–2024. Combining longitudinal scientometric analysis with counterfactual modeling, we identify a major structural break in Iran's research trajectory after 1979. Iran experienced a substantial post-2000 recovery in publication *volume*, but this recovery was accompanied by a persistent *quality* gap, with Field-Weighted Citation Impact (FWCI) and top-cited-paper shares remaining below those of several comparator systems. To assess the long-run opportunity cost of the disruption, we use two complementary counterfactual strategies: a comparison-based Synthetic Control Method (SCM) and a set of developmental-growth proxy scenarios. Across specifications, the qualitative conclusion is stable: the post-1979 interruption is associated with a durable loss of scientific compounding capacity. At the same time, the quantitative magnitude of the cumulative shortfall is sensitive to counterfactual specification, indicating that no single benchmark should be treated as a definitive historical forecast. We therefore interpret the SCM results as comparison-based benchmarks and the developmental-growth proxies as illustrative upper- and mid-range scenarios rather than literal one-to-one counterfactuals. We interpret these findings through the lens of the *Middle-Income Trap* (MIT) and the political economy of the *Rentier State*. We argue that Iran's post-revolutionary science policy contributed to a quantity–quality paradox in which expansion in scientific output was not matched by comparable gains in global scientific influence, while also recognizing that observed citation-impact measures are shaped by international collaboration constraints and bibliometric coverage limitations. The findings highlight the long-run developmental cost of political disruption while emphasizing that sustained scientific catch-up depends not only on output growth, but also on institutional continuity, international integration, and incentive structures that reward impact rather than volume alone.

**Data availability statement:** We have deposited the minimal dataset, analysis scripts, and supporting files necessary to reproduce the study findings in Zenodo. The materials are publicly available without restriction at: https://doi.org/10.5281/zenodo.20180908 The same reproducibility package is also provided as Supporting Information file S1_Data.zip and is available in the public GitHub repository: https://github.com/Ehsan-Roohi/Plos_One.

**Funding:** The author(s) received no specific funding for this work.

**Competing interests:** The authors have declared that no competing interests exist.

## 1. Introduction

In the 1960s and 1970s, Pahlavi's Iran was defined by an ambitious, state-led modernization drive. Central to this national project, often articulated through development plans and concepts like the "Great Civilization" (Tammadon/e Bozorg), was the construction of a modern, Western-aligned scientific and higher education infrastructure. This was a deliberate national development strategy [1]. It fueled the establishment and rapid expansion of elite institutions such as Pahlavi University in Shiraz, designed to be a national beacon, and the Aryamehr University of Technology (now Sharif), intended to train the engineering and technical cadre for a nascent industrial economy. This strategy was predicated on the belief that scientific capacity was synonymous with national sovereignty and long-term economic prosperity [2].

The momentum of this pre-revolutionary push was not merely rhetorical. Bibliometric evidence from the period points to a nascent scientific system experiencing rapid expansion. By the late 1970s, Iran had reached a meaningful takeoff point in publication output and institutional capacity. In comparative terms, its scholarly production was broadly competitive in aggregate scale with several middle-income and newly industrializing systems, including South Korea, Taiwan, and Turkey. This pre-1979 pattern establishes an important baseline for the analysis that follows: Iran entered the revolutionary period not as a marginal scientific system, but as one with clear upward momentum and plausible developmental potential.

The 1979 Iranian Revolution, however, fundamentally altered this path. While all revolutions reshape national institutions, the impact on Iran's scientific enterprise was immediate, profound, and structural. The primary shock was the "Cultural Revolution" (Enqelab-e Farhangi), launched in 1980. With the stated goal of "Islamizing" universities and purging secular and Western influences, the policy resulted in the complete closure of all higher education institutions from 1980 to 1983. This three-year shutdown was not a pause; it was a systemic reset that severely reduced the nation's human capital. This event was the first and most direct cause of the collapse in Iran's scientific growth. As a direct result of the purges, ideological vetting, and forced exoduses that accompanied this period, the number of university teaching staff fell sharply. Drawing on data attributed to the Ministry of Culture and Higher Education and reproduced in multiple secondary sources, the total number of university teaching staff (including full-time, part-time, and sessional staff) declined from 16,877 in the 1979–80 academic year to 9,042 by 1982–83, implying a loss of 7,835 positions [3,4]. The same statistical series also reports a decline from 2,455–1,424 in female teaching staff and from 14,422–7,618 in male teaching staff over the same interval [4]. Because the available historical series report aggregate teaching staff rather than a clean separation of research-active full-time faculty, we interpret this contraction as a proxy for a major loss of university teaching and research capacity rather than as an exact count of active researchers alone.

This internal, structural shock to the academic system was immediately exacerbated by a profound external crisis: the outbreak of the Iran-Iraq War (1980–1988). The war diverted all national resources and focus toward military defense and national survival. This paper provides a quantitative analysis of this combination

of shocks: university closures, the loss of half the nation's professoriate, total war, and international isolation. Historical evidence suggests that together these shocks triggered a sharp decline in scientific productivity. We empirically investigate the period from 1980 through the late 1990s as Iran's prolonged post-revolutionary stagnation phase. This phase began with the sharp collapse of the 1980s and extended into the slow and incomplete recovery of the 1990s. Our aim is to test whether this broader interval constituted a period of deep disconnection from the global scientific community. In this revised terminology, the "lost decade" refers more narrowly to the 1980s collapse phase, while the 1990s are interpreted as a slow transition out of the trough rather than as a continuation of the deepest contraction.

Iran's subsequent journey—an initial collapse, a long stagnation, and then a rapid but incomplete recovery from the late 1990s onward [5–8]—does not fit neatly into the established models of scientific system recovery. Historical analysis reveals at least two distinct archetypes of recovery from systemic political shocks. The first can be described as a model of prolonged stagnation followed by slow, policy-driven reconstruction. The collapse of the Soviet Union provides a significant example: its scientific enterprise experienced a long period of institutional stress, fiscal decline, and delayed rebuilding, with recovery only emerging later under new policy arrangements [9]. At the same time, the post-Soviet case differs fundamentally from Iran's in one crucial respect: it involved the dissolution and fragmentation of a super-state and its research system, whereas post-revolutionary Iran retained a centralized national state but underwent an internally driven political-ideological rupture. For this reason, the Soviet case is used here as a broad contrast in recovery dynamics rather than as a directly comparable institutional analogue. The second archetype is one of a near-total reset followed by rapid, state-led exponential growth. China's trajectory after the Cultural Revolution (1966–1976) is a prominent example of this pattern. During that period, scientific activity was severely disrupted, and publication output in major international outlets fell to extremely low levels. After 1978, however, China's recovery was supported by structural conditions that sharply distinguish it from Iran's later experience: reintegration into the global economy, large-scale state investment in science and technology, expanding industrial export capacity, and the absence of sustained international sanctions comparable to those faced by Iran. For this reason, China is not treated in this paper as a theoretically equivalent recovery model for Iran. Rather, it serves as a contrasting example of how a politically disrupted scientific system can recover under far more favorable conditions of global integration, fiscal scale, and industrial transformation [10].

While historical narratives have captured the political dimensions of the revolution and the war, the long-run opportunity cost of this disruption for Iran's scientific development remains insufficiently quantified. This paper situates Iran's interrupted trajectory within a comparative scientometric framework and asks two related questions. First, what was the immediate and long-term impact of the 1979 Revolution on Iran's scientific publication trajectory relative to comparable systems? Second, how large was the long-run loss in scientific compounding capacity associated with the post-1979 interruption when evaluated against multiple counterfactual benchmarks? Framed in this way, the study is not limited to estimating a single missing-output number; it also examines how the magnitude of the inferred loss depends on the counterfactual logic being used.

We address the second question by developing a set of counterfactual exercises rather than relying on a single benchmark. These include a comparison-based Synthetic Control Method (SCM) and a series of developmental-growth proxy scenarios designed to bracket the plausible range of long-run divergence. While scientific output is an imperfect measure of national innovation capacity, it remains a valuable indicator of knowledge accumulation, institutional continuity, and human-capital development [2]. By systematically comparing Iran's observed trajectory with multiple counterfactual paths, this paper provides a quantitative lens on the long-term developmental consequences of the 1979 disruption while remaining attentive to uncertainty in counterfactual construction. More broadly, it seeks to explain how political shock, institutional interruption, and subsequent incentive structures can reshape the long-run dynamics of scientific recovery.

A central contribution of the paper is therefore methodological as well as substantive. Rather than treating any single comparator country as a literal historical substitute for Iran, we distinguish between different counterfactual roles: conservative comparison-based benchmarks, broader donor-pool reconstructions, and developmental-growth proxy scenarios.

This distinction is important because the qualitative conclusion of post-1979 divergence is robust across approaches, whereas the quantitative magnitude of the cumulative shortfall varies with benchmark choice. The theoretical discussion that follows provides the framework for interpreting this divergence not simply as a temporary decline in output, but as a longer-run reconfiguration of the relationship between scientific scale, quality, and state structure.

## 2. Theoretical framework: The Middle-Income trap and rentier science

To interpret the quantitative divergence observed in the bibliometric data, this study integrates two theoretical frameworks: the *Middle-Income Trap* (MIT) in national innovation systems and the political economy of the *Rentier State*.

### 2.1. The Scientific Middle-Income trap

The Middle-Income Trap describes a developmental stage where countries successfully transition from low-income to middle-income status via factor accumulation (e.g., building universities, increasing literacy) but fail to transition to high-income status due to an inability to shift toward productivity-driven innovation [11]. Applied to science policy, this manifests as the "quantity-quality paradox": a nation achieves high volumes of scientific output (accumulation) but fails to achieve global scientific impact or industrial relevance (innovation).

In the scientometric context used here, a "scientific middle-income trap" is defined operationally as a condition in which a national research system succeeds in expanding publication scale over time but exhibits persistently weaker conversion of that scale into normalized scientific influence. In empirical terms, the key symptom is not low output per se, but a decoupling between growth in publication volume and growth in field-normalized impact indicators such as Field-Weighted Citation Impact (FWCI). Under this interpretation, the trap is expressed as prolonged accumulation without commensurate transition to sustained high-impact performance.

As illustrated in Fig 1, Iran appears to have advanced substantially through the accumulation phase of scientific development. However, movement from accumulation to innovation typically requires institutional arrangements that reward meritocracy, protect intellectual property, and facilitate the circulation of tacit knowledge [12]. In this framework, Iran's post-revolutionary trajectory is best interpreted as a case of partial advancement in scale without a corresponding transition to consistently high-impact innovation.

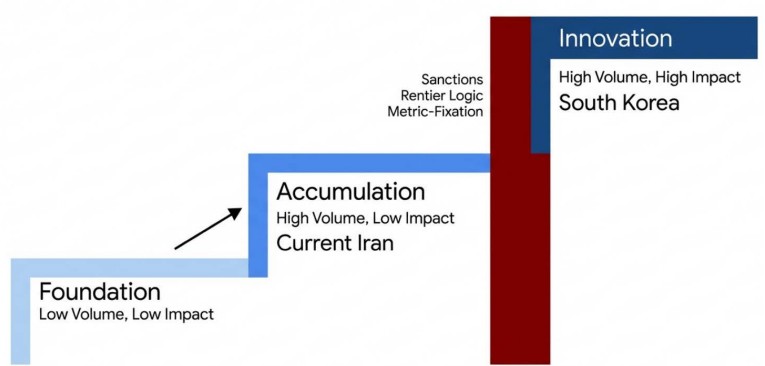

**Fig 1. The Scientific Middle-Income Trap: Conceptual Interpretation of Iran's Trajectory.** Stylized mapping of Iran's scientific development onto a capabilities-escalation framework. The figure is intended as an interpretive schematic rather than a direct empirical estimate. It illustrates the argument that Iran achieved substantial expansion in scientific scale, but that this expansion was not matched by a comparable transition to sustained innovation capacity and global research impact.

## 2.2. Rentier vs. Developmental state logic

A useful way to interpret Iran's post-1979 scientific divergence is through the contrast between rentier-state and developmental-state logics. In developmental states, scientific expansion is more likely to be tied to industrial upgrading, export competitiveness, and long-horizon capability building. South Korea is a prominent example of this model: as a resource-poor and security-pressured state, it developed institutions in which science and technology policy were closely linked to industrial performance and export discipline [13].

Iran, by contrast, is more plausibly interpreted through the political economy of the *Rentier State*, in which public revenue depends heavily on external rents rather than domestic taxation. In such systems, science policy can more easily acquire distributive, bureaucratic, or prestige-oriented functions in addition to productive ones [14]. This institutional logic may generate volatility in funding priorities, weaker links between research and industrial upgrading, and stronger incentives for visible expansion in scale than for long-run gains in research quality and innovation capacity. At the same time, the rentier-state interpretation should not be treated as temporally uniform across the entire post-2000 period. Iran's oil economy changed substantially in the 2010s and 2020s under the combined effects of sanctions, revenue compression, and repeated attempts at fiscal adjustment and diversification. For this reason, the rentier argument in this paper is not that science funding simply rose or fell mechanically with oil income in a linear way. Rather, the claim is that a rent-dependent institutional structure can continue to shape the volatility, short-horizon allocation logic, and weak industrial anchoring of science policy even when oil rents themselves become more constrained or unstable.

Within this framework, the long-run divergence documented in this paper should not be interpreted as the mechanical product of a single political shock alone. Rather, the post-1979 interruption interacted with a broader institutional setting in which scientific scale could expand without the same degree of export-discipline, industrial anchoring, and international integration observed in developmental-state cases. For this reason, South Korea is most useful here as an illustrative developmental benchmark, not as a literal one-to-one historical substitute for Iran.

This MIT-based interpretation does not replace national innovation systems theory; rather, it provides a more specific lens for interpreting one recurrent pattern within it. National innovation systems theory helps explain how institutions, policy coordination, and industrial linkages shape scientific performance, whereas the MIT framing highlights the particular developmental problem of remaining stuck in a scale-expansion phase without comparable upgrading in impact and innovation quality.

More generally, the argument developed in this paper is not that all political disruptions generate identical scientometric consequences. Different forms of disruption—revolutionary rupture, war, regime collapse, state dissolution, or externally imposed transition—operate through different mechanisms and leave different institutional legacies. The Iranian case is therefore interpreted as a historically specific combination of ideological rupture, university closure, war mobilization, and post-disruption incentive restructuring, rather than as a universal model of how political shocks affect science.

## 3. Data and methods

### 3.1. Data source and query

The primary data source for this study is the Elsevier Scopus database, a comprehensive index of scholarly publications. We collected annual publication counts (limited to documents classified as "articles" or "reviews") for the period 1960–2024 for Iran and a carefully selected peer group.

This comparative group was not chosen arbitrarily. It was assembled to benchmark Iran's trajectory against several distinct families of scientific development rather than against a single presumed historical substitute. The comparison set includes:

- Developmental-state benchmarks (South Korea, Taiwan, Singapore): These cases represent high-growth, export-oriented systems in which science and technology policy became closely linked to long-run industrial upgrading.

- Stable regional and mature scientific systems (Israel, Netherlands): Israel serves as a benchmark for a stable, high-performing regional scientific power, while the Netherlands represents a mature and internationally integrated research system.

- Additional comparative trajectories (Greece, China): Greece offers a European comparison with a different growth and integration pattern, while China provides a contrast case of political disruption followed by large-scale state-led recovery under conditions of post-1978 global economic integration, expanding industrial capacity, and unusually strong state investment in research. It is included here as a comparative trajectory rather than as a substantively equivalent recovery model for Iran.

The resulting annual publication counts form the basis of our longitudinal time-series analysis. Throughout the paper, we distinguish between the annual output gap and the cumulative shortfall relative to a given benchmark. Unless otherwise stated, cumulative shortfall estimates are reported as benchmark-specific and may vary modestly across model summaries depending on donor specification, terminal-year convention, and rounding. For this reason, the manuscript does not treat any single cumulative publication-loss figure as a definitive historical estimate. The annual gap refers to the year-by-year difference between Iran's observed output and a specified counterfactual path, whereas the cumulative shortfall refers to the sum of those gaps over time up to the terminal year of analysis. We retain the term "knowledge deficit" as a shorthand interpretive label in the discussion, but the underlying quantities remain benchmark-dependent. For robustness purposes, we also estimated an expanded-donor SCM specification in which Malaysia, Thailand, and Mexico were added to the core donor pool. These countries were not part of the paper's primary comparative framing, but were introduced in a secondary robustness exercise to test whether the SCM results depended on a narrow donor set. Spain was considered but excluded from the expanded-donor robustness analysis because its pre-1979 predictor coverage was insufficient for a balanced implementation. For clarity, the term "knowledge deficit" is used in this paper only to refer to the *cumulative shortfall* in publication output relative to a specified counterfactual benchmark over a defined period. By contrast, the *annual gap* refers to the year-specific difference between Iran's observed output and the benchmark in a given year. Unless explicitly stated otherwise, references to "knowledge deficit" in the manuscript should therefore be read as cumulative rather than annual.

### 3.2. Analytical approach and metrics

Our study employs a multi-dimensional scientometric approach to provide a comprehensive assessment of Iran's scientific trajectory. The analysis integrates both quantitative and qualitative metrics to move beyond simple publication counts and capture a more nuanced picture of scientific development. The methodology is structured as follows:

1. Longitudinal Analysis of Publication Output: We first analyze the growth trajectories of all eight countries in terms of publication volume across three distinct historical periods: 1960–1979 (Foundation), 1980–1999 (Divergence), and 2000–2024 (Recovery and New Order). This establishes the scale of the divergence.

2. Analysis of Scientific Impact and Quality: To assess the quality and influence of the research, we incorporate two key normalized metrics from SciVal for the period 1996–2024:

   - Field-Weighted Citation Impact (FWCI): To assess research quality and influence, we used the Field-Weighted Citation Impact (FWCI). This standard metric measures citation impact relative to the global average in the same field.

   - Outputs in Top 10% Percentiles: As a supplementary indicator of highly cited research, we analyzed the SciVal metric "Outputs in Top 10% Percentiles," which represents the share of a country's papers that fall within the top 10% most cited outputs in the indexed data universe. In the revised interpretation adopted here, this metric is used descriptively and is interpreted more cautiously than FWCI in cross-country comparisons, because disciplinary portfolio differences may affect direct comparisons across national systems.

This dual analysis of quantity and quality allows for a robust comparison of national scientific systems.

3. Counterfactual Modeling: Finally, we develop and compare a suite of counterfactual exercises to assess the scale of post-1979 divergence under alternative benchmark logics. Rather than treating any single counterfactual path as definitive, we use these exercises to evaluate the robustness of the qualitative conclusion and the sensitivity of the cumulative shortfall to model specification.

As an additional robustness check motivated by disciplinary-composition concerns, we also examined field-specific FWCI trends for six major subject areas in the later SciVal period. These field-level comparisons are used to assess whether the observed cross-country quality gap persists after subject-level normalization.

For the baseline SCM specification, we retained four World Bank predictors: GDP per capita, tertiary enrollment, government expenditure, and trade openness. The rationale for this choice is primarily historical coverage. Because the SCM treatment year is 1979, the baseline model must reproduce Iran's pre-treatment profile using predictors that are available with sufficient consistency across the donor pool before 1979. In practice, these four variables provided the most usable and comparatively balanced long-run coverage for the pre-treatment period. By contrast, conceptually relevant variables such as R&D expenditure as a share of GDP and researchers per capita are only sparsely available for most donor countries in the pre-1979 period, making them unsuitable for inclusion in the baseline historical SCM without substantially shrinking the donor pool or introducing severe missing-data imbalance. International scientific collaboration rates were also not included in the baseline SCM because they are better interpreted as bibliometric integration outcomes than as core macro-structural predictors in a long-horizon pre-treatment matching design.

The counterfactual exercises in this study therefore play different roles. The SCM specifications are used as comparison-based benchmarks constructed from pre-treatment fit, while the developmental-growth proxy exercises are used as scenario-based sensitivity analyses that illustrate how the inferred long-run shortfall changes under alternative growth analogies. This distinction is central to the interpretation of the results: the SCM is not intended to identify a single "true" alternate history, and the developmental-growth proxies are not interpreted as literal forecasts. Instead, the combination of these approaches is used to separate qualitative robustness from quantitative benchmark sensitivity.

### 3.3. Operationalizing the Scientific MIT interpretation

To make the MIT interpretation more explicit, we distinguish between two dimensions of scientometric performance: (i) scale accumulation, measured by publication output, and (ii) impact conversion, measured by field-normalized influence indicators such as FWCI. In this framework, a scientific MIT-like pattern is one in which output expands strongly over time while normalized impact improves only weakly or remains persistently below the levels observed in more innovation-intensive comparator systems. This is not treated as a formal binary classification, but as an empirically assessable pattern of decoupling between quantity growth and impact upgrading.

As a simple empirical check for this interpretation, we examine the Iran-only relationship between publication growth and FWCI over the SciVal period (1996–2024). The purpose of this exercise is not to estimate a causal structural model, but to assess whether the observed trajectory is consistent with sustained accumulation accompanied by weaker impact conversion.

### 3.4. Counterfactual models

Our counterfactual analysis combines two distinct modeling strategies. The first is a comparison-based Synthetic Control Method (SCM), in which Iran is compared with a weighted combination of donor countries selected to reproduce its pre-1979 structural profile as closely as possible. The second is a set of developmental-growth proxy exercises, in which the post-1979 growth dynamics of comparator systems are applied to Iran's late-1970s baseline under alternative assumptions. The SCM and proxy exercises are therefore not treated as interchangeable estimates of one quantity; rather, they

are used to evaluate whether the post-1979 divergence remains qualitatively robust across counterfactual constructions and to assess how strongly the cumulative shortfall depends on benchmark choice. Additional details on the counterfactual models, donor-pool robustness checks, placebo tests, and benchmark-sensitivity analyses are provided in Appendix A, with full empirical results reported in Appendix A.1.

### 3.5. Limitations of the study

This study has several limitations that should guide interpretation. First, historical publication coverage is imperfect, especially before the mid-1990s, because Scopus indexing becomes more complete in later decades. Although this limitation likely affects all countries in the comparison set to some degree, early-period counts should still be interpreted as comparative trend indicators rather than exact censuses of scientific output.

Second, citation-based quality metrics such as FWCI should be interpreted cautiously in uneven global bibliometric systems. For countries facing sanctions, weaker integration into high-visibility collaboration networks, and partial publication in regional or non-Scopus-indexed venues, observed citation impact may reflect not only intrinsic research quality and domestic incentive structures, but also differences in visibility, collaboration opportunity, and database coverage.

Third, the counterfactual analysis is benchmark-sensitive. The developmental-growth proxy scenarios are not literal alternate histories, and the SCM results remain sensitive to donor-pool composition, predictor availability, and pre-treatment fit. This is especially important in specifications where China receives a dominant donor weight: such outcomes should be interpreted as optimization results rather than as claims of deep historical or theoretical equivalence. More broadly, China is informative here as a contrast case of recovery after political disruption, but its post-1978 scientific expansion occurred under structural conditions that differ substantially from Iran's, including deeper global integration, larger state R&D mobilization, and the absence of comparable sanctions.

Fourth, some of the broader institutional interpretations in the paper are supported more as pattern-based explanations than as direct causal tests. In particular, the scientific MIT interpretation is operationalized through the decoupling between scale growth and normalized impact, but it is not presented as a deterministic binary classification. Similarly, the rentier-state argument should be read as an institutional-political interpretation of the long-run allocation environment rather than as a direct econometric demonstration linking annual oil-rent fluctuations to annual science-budget changes in the 2010s and 2020s.

Finally, some historical indicators are necessarily approximate. The university staffing contraction documented for the early post-revolutionary period refers to aggregate teaching staff rather than a clean census of research-active faculty alone, and the precise quantitative magnitude of the cumulative publication shortfall varies across benchmark families. For this reason, the paper places greater weight on the robustness of the qualitative conclusion—namely, the persistence of post-1979 divergence—than on any single point estimate.

## 4. Results: A Tale of Two Trajectories – Quantity and Quality

Figs 2–5 present the annual trajectories of Scopus-indexed publications for Iran and selected comparator systems over 1960–2024. These figures provide the descriptive foundation for the subsequent analysis of divergence in scientific scale, recovery, and longer-run compounding dynamics.

Our analysis reveals a story of divergence that unfolds not only in the volume of scientific output but also in its quality and international impact. We integrate publication counts with normalized citation and impact indicators to provide a multidimensional view of Iran's scientific trajectory in comparison with selected developmental, regional, and mature scientific systems.

### 4.1. 1960–1979: The Era of Iran's Scientific Lead

This period establishes the baseline from which the later divergence emerged. Iran was not a lagging scientific system before 1979; rather, it was a rapidly expanding one with visible upward momentum. In the Scopus-indexed series used

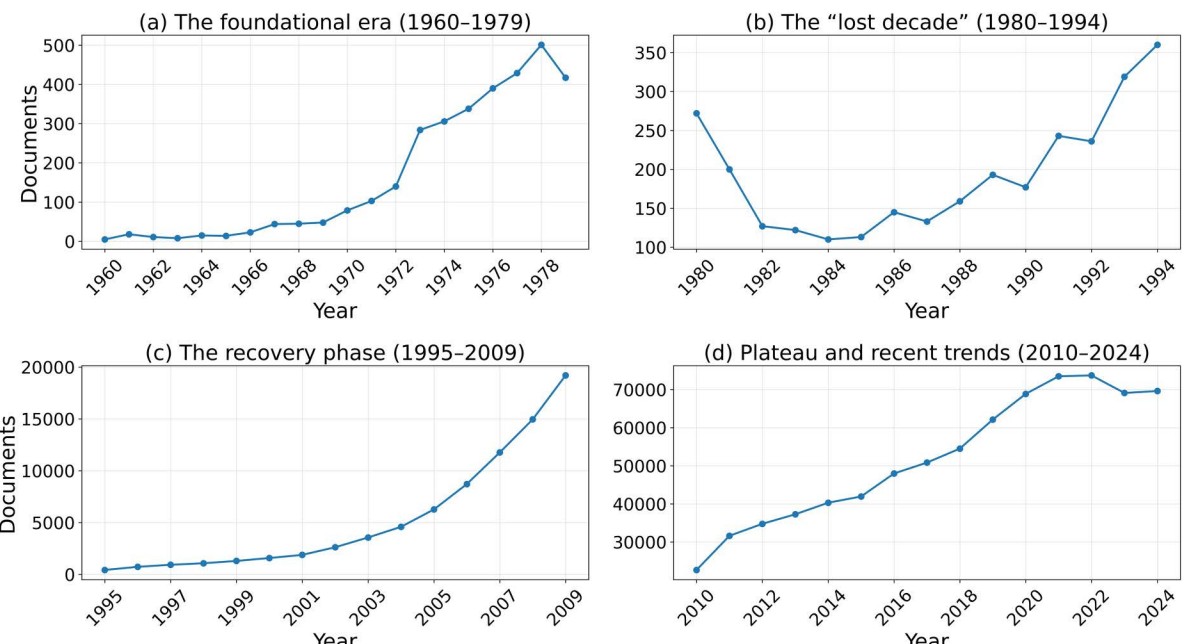

**Fig 2. Iran's publication output across four distinct periods, illustrating the pre-revolutionary growth, post-revolutionary collapse, subsequent recovery, and the recent plateau.**

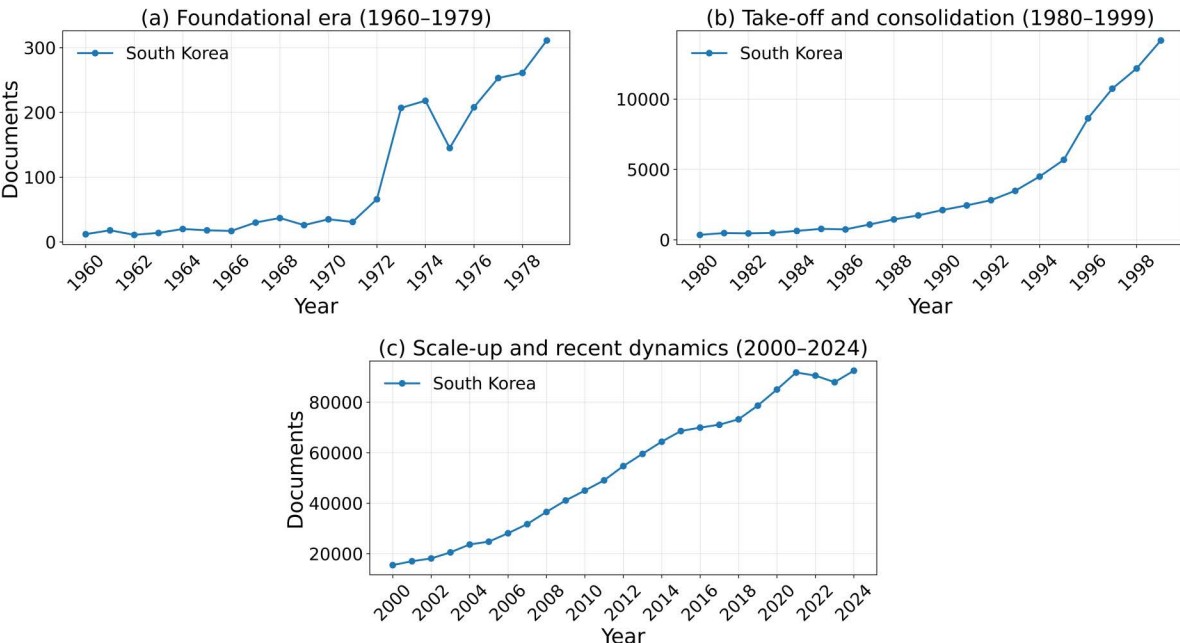

**Fig 3. Annual Scopus-indexed articles and reviews for South Korea (1960–2024) across three periods.**

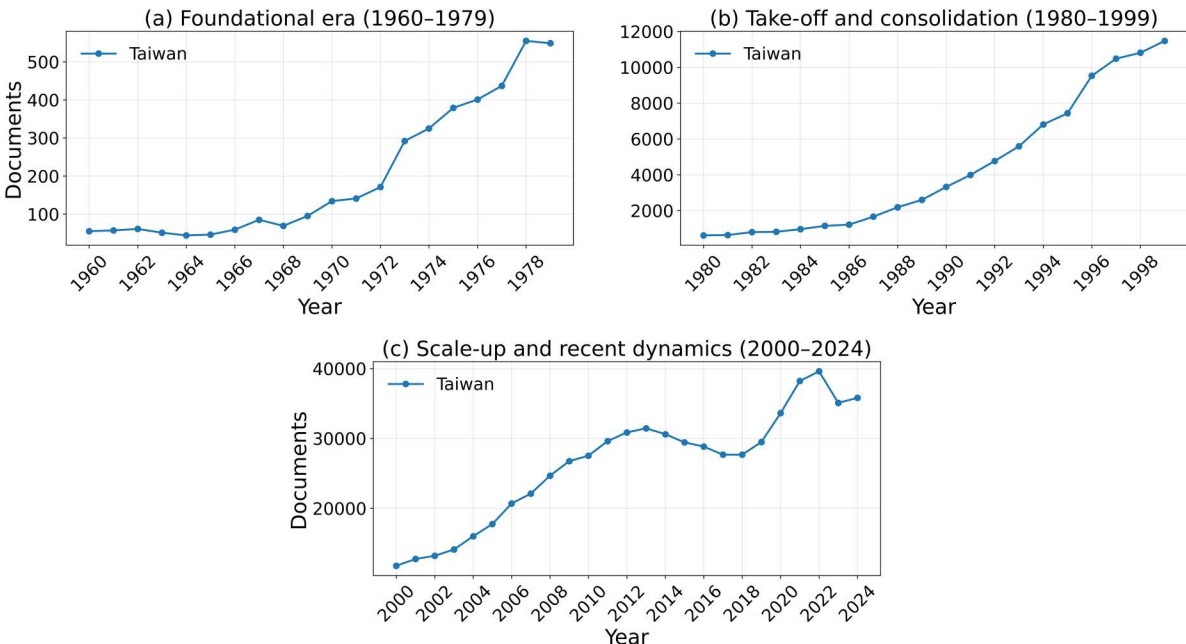

**Fig 4. Annual Scopus-indexed articles and reviews for Taiwan (1960–1979, 1980–1999, and 2000–2024).**

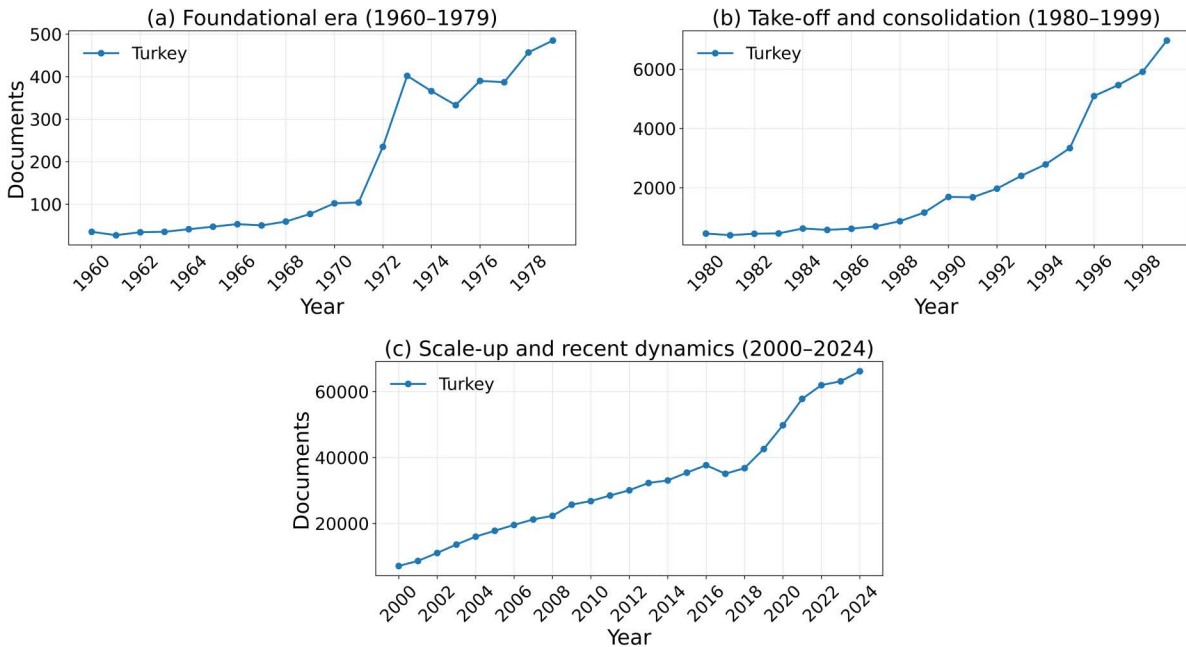

**Fig 5. Annual Scopus-indexed articles and reviews for Turkey (1960–1979, 1980–1999, and 2000–2024).**

for the main longitudinal comparison, Iran's publication output increased substantially during the 1970s, reaching approximately 500 papers in 1978. In absolute output, this level exceeded South Korea and was in the same broad range as Taiwan and Turkey. However, this comparison should be interpreted cautiously, because it refers to aggregate publication counts rather than population-normalized scientific intensity. A simple 1978 per-capita check shows that Iran remained ahead of South Korea on this metric, but substantially below Taiwan (Using the 1978 publication counts in the Scopus-indexed series and approximate 1978 population totals, the resulting publications-per-million values are about 30.8 for Taiwan, 13.9 for Iran, 10.6 for Turkey, and 7.1 for South Korea.). For this reason, the pre-1979 evidence is interpreted here as showing that Iran had become regionally competitive in scientific scale and growth momentum, rather than that it uniformly outperformed all comparator systems on a per-capita basis.

### 4.2. 1980–1999: Collapse, stagnation, and partial recovery

This period marks the clearest phase of divergence, but it was not internally uniform. The 1980s constituted the sharp collapse phase: Iran's publication output fell steeply after the Revolution and reached a trough of roughly 122 papers in 1984. The 1990s, by contrast, were better characterized as a slow and incomplete recovery phase rather than as a continuation of the initial collapse. Although Iran did not surpass its 1978 publication level until the mid-1990s, that threshold crossing is itself an important inflection point: it marks the end of the immediate post-revolutionary trough, but not the restoration of the pre-1979 growth regime. In that sense, the 1990s represent transition rather than full recovery. The broader interruption from 1980 through the late 1990s therefore reflects a sequence of collapse, stagnation, and only partial re-entry into sustained scientific expansion.

By contrast, several comparator systems entered sustained expansionary regimes during the same period. South Korea and Taiwan moved decisively into long-run takeoff, while China began a major upward turn in the 1990s and Turkey also expanded substantially. By the end of the century, the gap between Iran and these systems was no longer a modest separation in level, but a structural divergence in growth regime. This divergence was not only quantitative: comparator systems were becoming more deeply integrated into international scientific networks, whereas Iran remained much more weakly connected to them.

### 4.3. 2000–2024: Recovery in scale, lag in impact

The stronger acceleration visible after 2000 should therefore be distinguished from the threshold recovery of the mid-1990s: the latter marks a return to the 1978 output level, whereas the former marks the beginning of a genuinely new scale-expansion phase. In the 21st century, Iran experienced a remarkable recovery in scientific scale. Annual publication output rose from roughly 2,500 papers in 2000 to a peak of around 80,000 in the early 2020s, reflecting major expansion in higher education and research capacity. This recovery is one of the most important descriptive findings of the paper: Iran did not remain in permanent stagnation, but rebuilt a large-scale publication system.

At the same time, recovery in quantity did not produce full convergence in scientific impact. Comparator systems followed different trajectories: some, such as South Korea, maintained both high output and stronger field-normalized impact; others, such as Taiwan and Turkey, remained important benchmarks for regional and developmental comparison; and large systems such as China moved onto a much steeper long-run scale trajectory. Iran therefore re-emerged as a high-volume scientific producer, but not as a fully converged high-impact system. The post-2000 pattern is best described as recovery in scale under continued divergence in influence.

### 4.4. Pre-Scopus Historical Reconstruction (1960–1995)

To contextualize the Scopus-based results, Figs 6–8 extend the analysis back to 1960 using Crossref's metadata and citation-to-date records. Because Scopus coverage before the mid-1990s is incomplete, Crossref provides a valuable

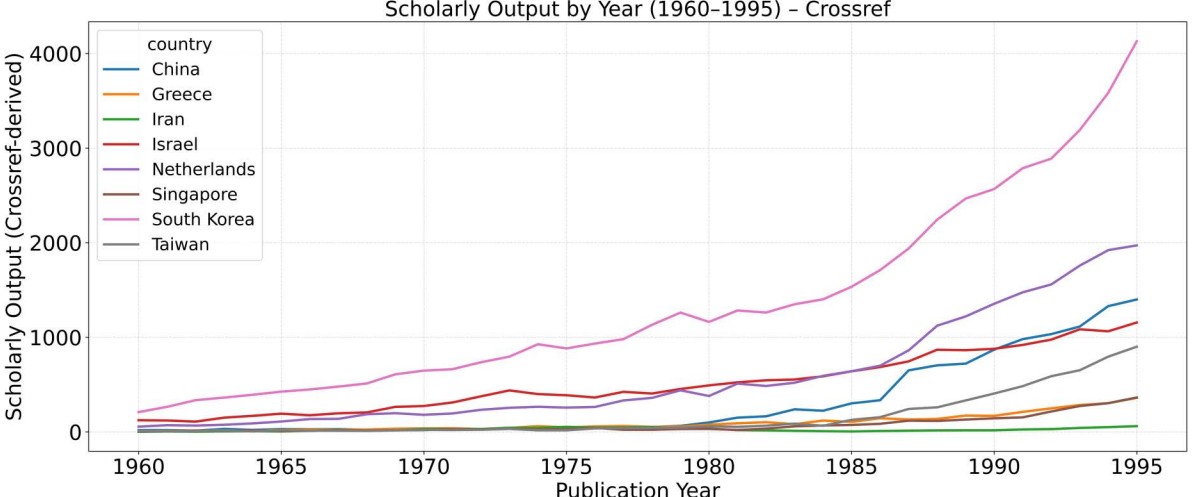

**Fig 6. Annual scholarly output (1960–1995) based on Crossref data.** These series are used as historical approximations for pre-Scopus trend reconstruction and are not expected to match the exact annual levels in the main Scopus-indexed dataset.

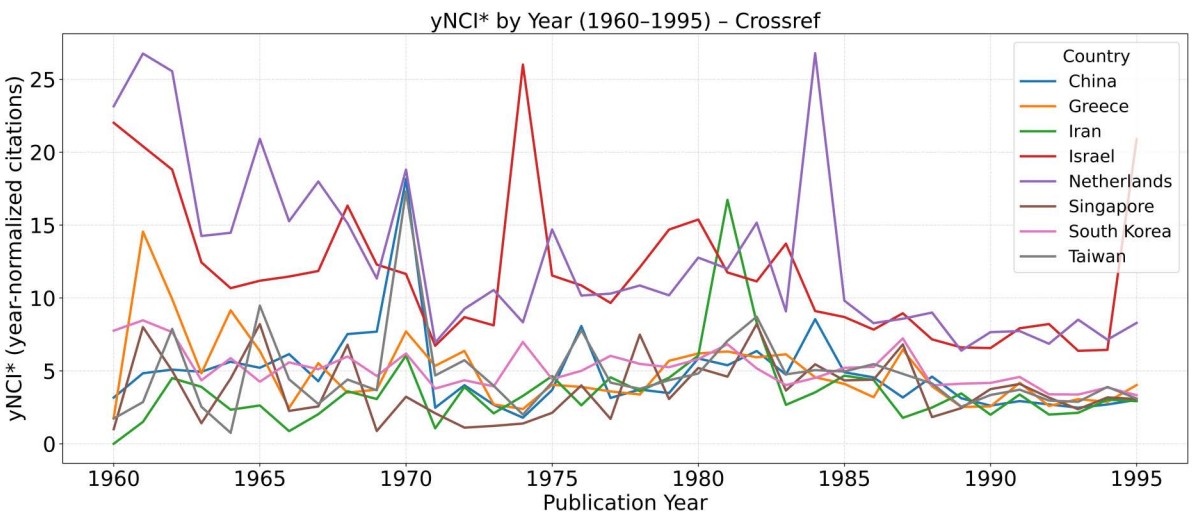

**Fig 7. Approximate year-normalized citation index (yNCI*) for 1960–1995, reconstructed from Crossref citation records.**

approximation of earlier publication and citation trends. These Crossref-based reconstructions are intended to recover broad historical patterns rather than to reproduce the exact annual counts observed in the main Scopus-indexed series. Accordingly, pre-1996 levels in the Crossref figures should not be read as numerically identical to the Scopus counts reported elsewhere in the paper.

These reconstructed metrics—denoted by an asterisk (*yNCI\**, *Top-10%\**)—use total citation counts to date rather than fixed five-year windows, offering a reasonable historical baseline for comparing national trajectories prior to the 1979 Revolution. The patterns corroborate the main narrative: Iran's pre-1979 scientific system exhibited healthy compounding growth and moderate global visibility, both of which collapsed abruptly during the 1980s before partial recovery in the 1990s.

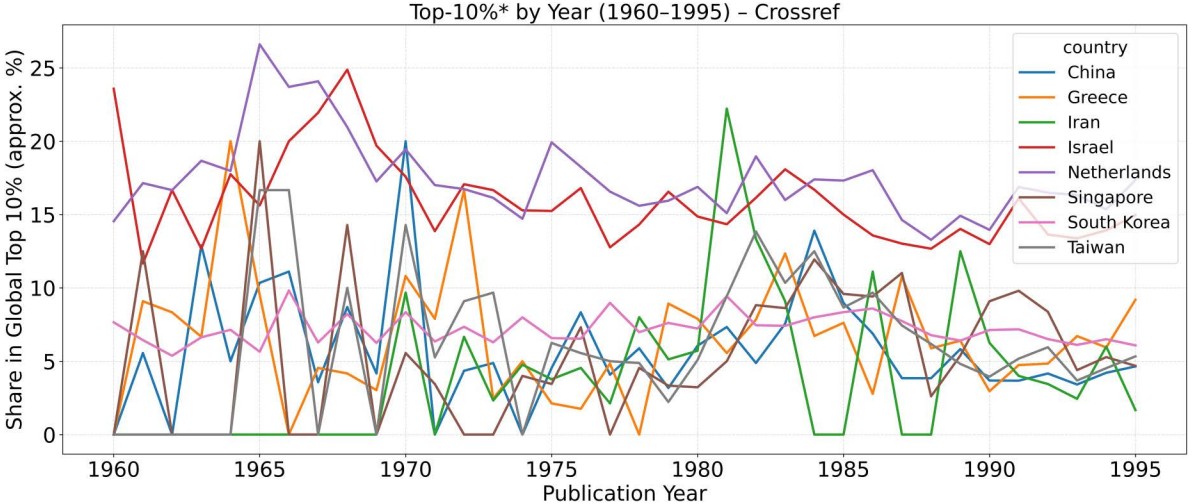

**Fig 8. Share of publications in the approximate global top 10%\* citation percentile (Top-10%\*) between 1960–1995.**

### 4.5. The Paradox of Quantity vs. Quality

Figs 9–11 illustrate the central paradox of Iran's post-2000 recovery: a successful restoration of research *scale* (quantity) alongside a persistent *lag* in research *influence* (quality).

The combined trends in Fig 9 reveal that Iran's research output expanded sharply after the early 2000s, demonstrating a remarkable recovery in absolute numbers. However, this growth in scale obscures a significant quality deficit when analyzed with normalized metrics.

As shown in Fig 10, Iran's Field-Weighted Citation Impact (FWCI)—a measure of citation impact normalized for subject field, year, and document type—showed gradual improvement, rising from well below the world average to approach it (FWCI $\approx$ 1.0) by 2020. While a significant achievement, this stands in stark contrast to its high-performing peers. Elite scientific nations like the Netherlands and Singapore consistently maintained an FWCI above 1.5, meaning their research was, on average, 50 percent more cited than the global average. Even South Korea and Taiwan sustained FWCI levels above Iran's and generally around the 1.1–1.2 range.

Fig 11 shows the percentage of publications falling within the top 10 percent of cited papers in the indexed data universe. Iran's share rose from under 5 percent in the late 1990s to about 12.8 percent in the recent 2015–2023 average. This provides a useful descriptive indication of improved citation visibility at the upper end of the distribution. At the same time, because cross-country comparisons on this metric may be affected by differences in disciplinary portfolio, we interpret it here as supplementary evidence rather than as a stand-alone, discipline-neutral measure of research quality.

To make the MIT interpretation more empirically explicit, we also examined the Iran-only relationship between publication scale and normalized impact over the SciVal period. Between 1996 and 2024, Iran's annual publication output increased from 877 to 76,939 papers, while FWCI rose from 0.53 to 1.12, or by a factor of about 2.11. Simple trend fits show that both output and FWCI improved over time, but not at comparable substantive rates. The key point is that publication scale expanded far more rapidly than normalized impact: between 1996 and 2024, annual output increased by a factor of about 87.7, whereas FWCI increased by a factor of about 2.11. The time trend in log publication output is extremely strong ($R^2 \approx 0.914$), while the FWCI trend, although clearly positive ($R^2 \approx 0.872$), indicates a much slower upgrading of normalized citation influence than the scale expansion alone would suggest. Likewise, the association between FWCI and log output is positive but incomplete ($R^2 \approx 0.723$), indicating that large increases in publication scale

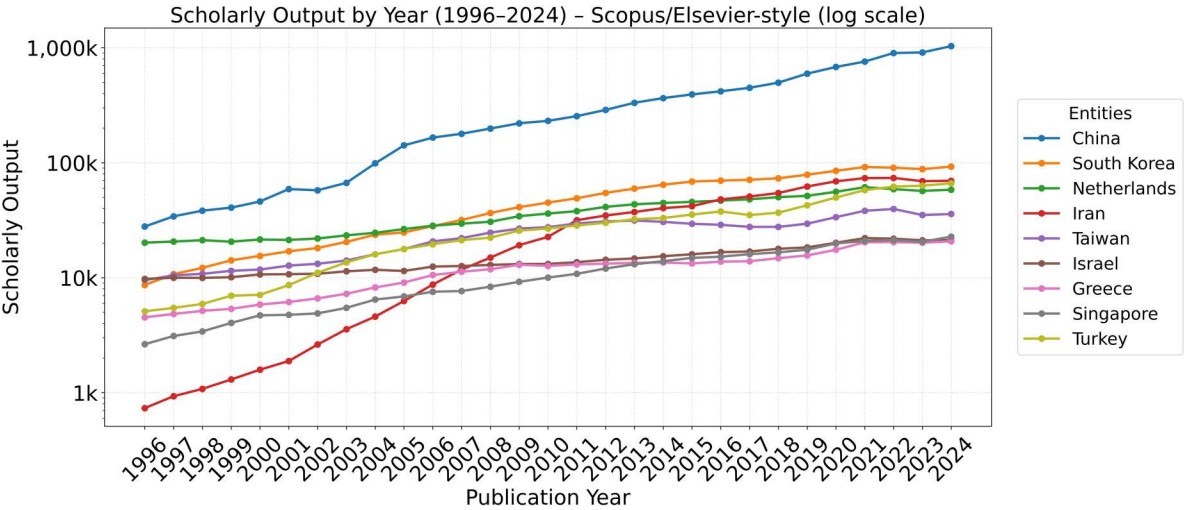

**Fig 9. Annual scholarly output of Iran and selected countries (1996–2024).**

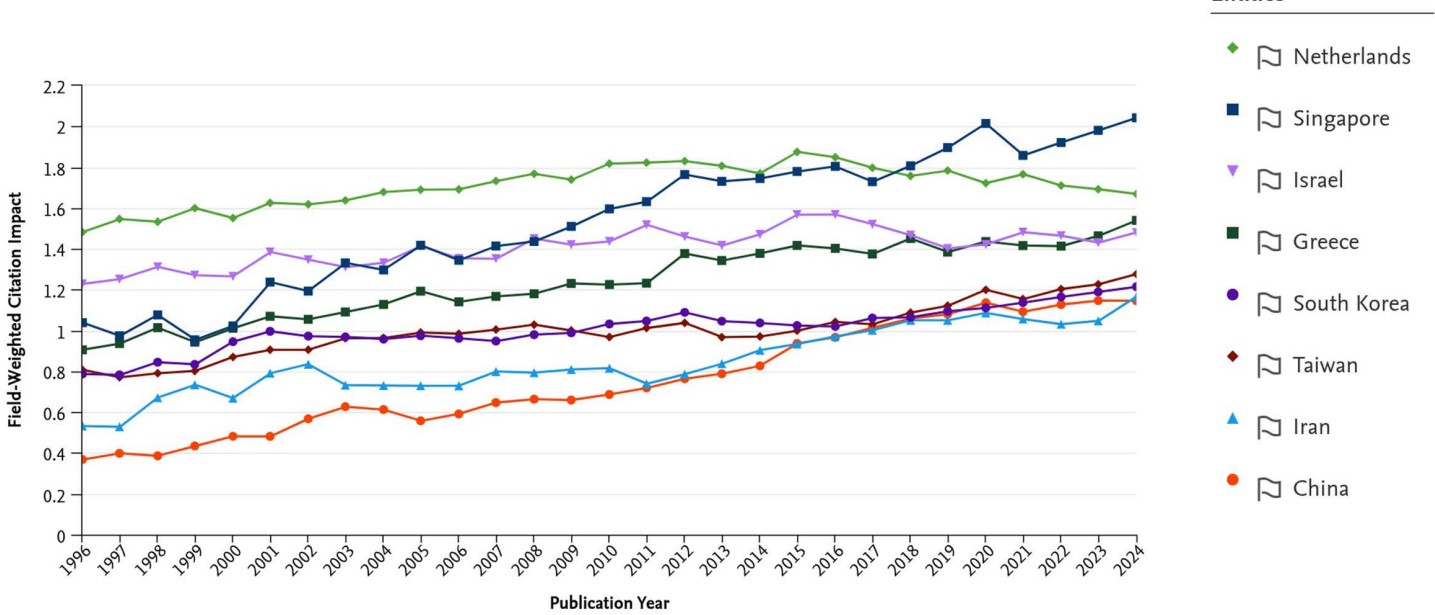

**Fig 10. Field-Weighted Citation Impact (FWCI) by publication year (1996–2024).**

were not matched by proportionate gains in normalized citation influence. We interpret this pattern as evidence consistent with a scientific MIT dynamic: successful accumulation in scale without a full transition to a high-impact innovation regime.

To further examine the concern regarding disciplinary composition, we conducted a field-specific robustness check using SciVal FWCI data for six major subject areas: Engineering, Medicine, Physics and Astronomy, Computer Science, Materials Science, and Mathematics. Because FWCI is already normalized by subject field, publication year, and document type, these comparisons provide a more discipline-sensitive benchmark than aggregate

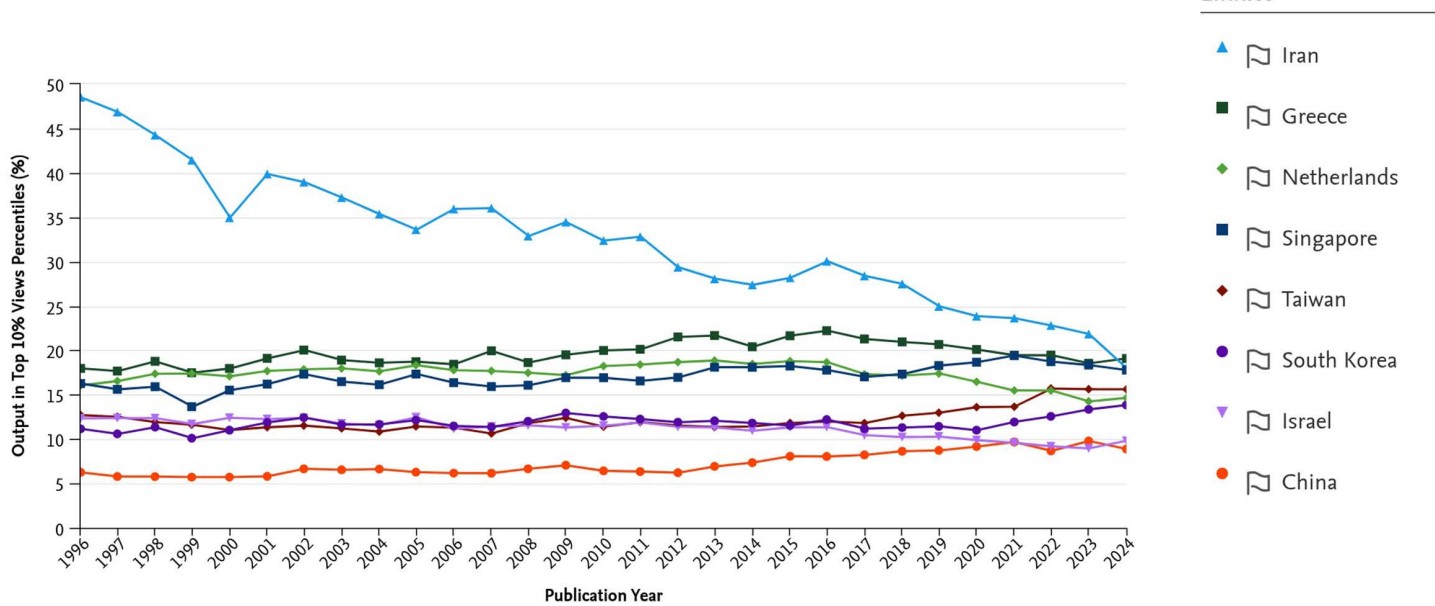

**Fig 11. Percentage of publications within the Top 10% citation percentiles (1996–2024).**

citation indicators alone. As shown in Fig 12, Iran's relative position varies across fields and is somewhat closer to selected comparator systems in areas such as Engineering and Materials Science. However, the broader pattern remains intact: substantial field-level gaps persist relative to the highest-performing and most internationally integrated systems, particularly the Netherlands and Singapore. This result strengthens the interpretation that the observed quality gap is not solely an artifact of national field mix, even though disciplinary composition still affects the interpretation of non-field-normalized elite-output indicators such as the top-10% citation share. Accordingly, the manuscript places greater interpretive weight on FWCI and the new field-specific robustness evidence than on the aggregate top-10% indicator alone.

This persistent gap between quantity and quality, while common in rapidly expanding systems, is particularly pronounced in the Iranian case. We interpret this lag as the result of multiple interacting structural factors operating at both the international and domestic levels, rather than as the product of any single mechanism alone.

Internationally, several mechanisms likely contribute to the observed citation-impact gap. Decades of relative isolation, exacerbated by geopolitical constraints and sanctions, have created structural limitations for Iranian science. This extends beyond travel or funding constraints and includes reduced opportunities for sustained collaboration with leading global research groups. As shown in Fig 13, Iran's share of internationally co-authored publications has generally remained below that of the most globally integrated comparator systems and only partially converged in the later years of the sample. Because international collaboration is a major driver of visibility and citation impact, reduced integration into elite research networks can mechanically depress observed FWCI even when domestic publication volume rises [15,16]. In addition, citation-based indicators derived from Scopus reflect the indexed publication universe and may underrepresent research published in regional or non-Scopus-indexed outlets. For this reason, the observed FWCI gap should not be interpreted as a pure measure of intrinsic research quality alone. As shown in Fig 14, changes in Iran's observed FWCI over time occurred alongside substantial variation in the share of internationally collaborative publications, reinforcing the need for a cautious interpretation of citation-impact trends.

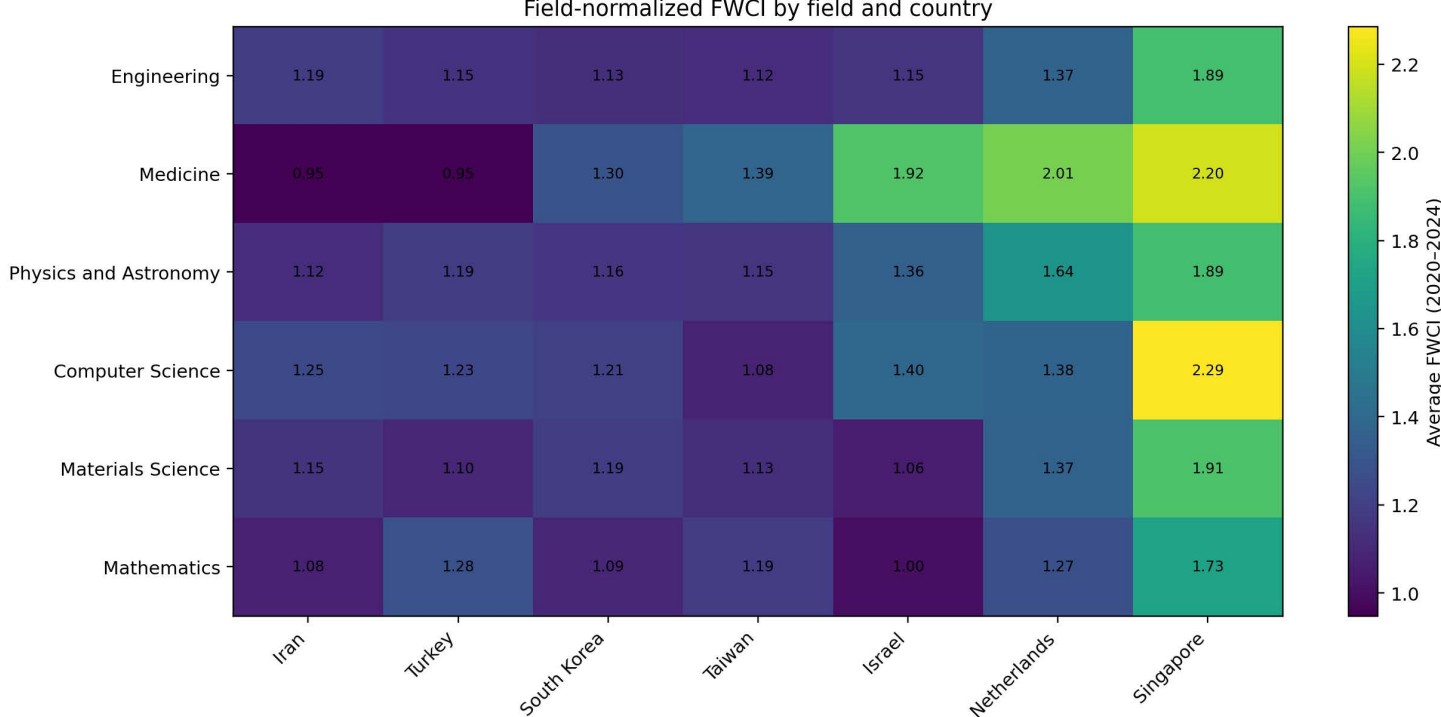

**Fig 12. Field-specific FWCI comparison across six major subject areas (average, 2020–2024).** To examine whether the observed cross-country quality gap is merely an artifact of disciplinary composition, we compare Field-Weighted Citation Impact (FWCI) across six major fields: Engineering, Medicine, Physics and Astronomy, Computer Science, Materials Science, and Mathematics. Because FWCI is field-normalized by design, these comparisons reduce the risk of conflating broad portfolio differences with research impact. The figure shows that although Iran is relatively closer to some comparator systems in selected fields, substantial gaps remain relative to the highest-performing systems, particularly the Netherlands and Singapore.

Domestically, one plausible contributing mechanism is the incentive structure associated with the post-1990s rebuilding strategy. The state-led push to restore scientific standing was strongly influenced by quantitative expansion and international ranking visibility, contributing to a science-policy and university-promotion system that often prioritized publication counts over longer-horizon impact. These institutional incentives may have encouraged volume-maximization strategies and incremental publication behavior. However, the present analysis does not identify the relative causal contribution of these domestic incentives as distinct from external constraints such as sanctions, export restrictions, and reduced access to international collaboration networks.

Taken together, these considerations suggest that the observed FWCI shortfall should be interpreted as a joint outcome of domestic incentive structures, reduced international collaboration opportunities, and the visibility constraints inherent in uneven bibliometric coverage. The manuscript therefore does not treat indexed citation impact as a fully bias-free measure of national scientific quality, but as a practically useful indicator whose interpretation requires institutional and geopolitical context.

Consequently, the policies that helped fuel the impressive recovery in scale may be understood as one part of a broader explanation for the persistence of the influence gap, which must also be interpreted in light of sanctions-related constraints, differential international integration, and bibliometric visibility. To synthesize these multi-dimensional findings, we present a comparative profile in Table 1.

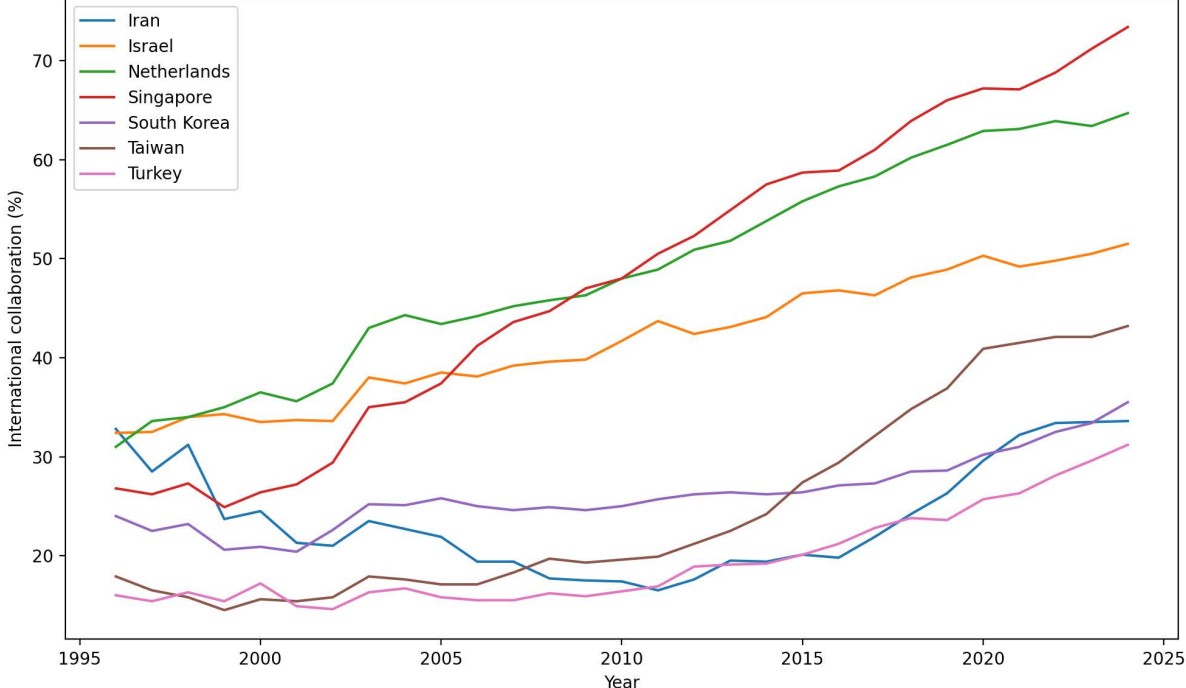

**Fig 13. International collaboration share by publication year (1996–2024).** Iran's share of internationally co-authored publications remains below that of highly integrated systems such as the Netherlands, Singapore, and Israel for most of the period, and below Taiwan in the later years. This pattern suggests that part of the observed citation-impact gap may reflect differential access to international collaboration networks and visibility channels, not domestic incentive structures alone.

### 4.6. Cross-country synthesis on a log scale

The comparative analysis in Fig 15, which juxtaposes trajectories on a logarithmic scale, provides the clearest visualization of the divergence. It reveals how small differences in growth *rates* (slopes), when sustained over decades, compound into massive, permanent differences in *level*. The vertical dashed line marks the 1979 breakpoint.

Across all panels, the data demonstrates that (i) Iran's pre-1979 slope was competitive, (ii) the post-1979 regime shift dramatically flattened this slope, raising its implied doubling time for roughly a decade and a half, and (iii) peers that maintained steep, low-doubling-time regimes (the Tigers; the Netherlands/Israel; later China) compounded into permanently higher levels. The resulting divergence is thus a consequence of *rate differentials sustained over long horizons*, not merely one-off level shocks.

This divergence in compounding growth is evident across all peer groups. In the comparison with the "Asian Tigers" (Panel A), Iran's pre-1979 momentum is clear; its slope is competitive—at times steeper than South Korea's—and its level is comparable to Taiwan's. Immediately after 1979, Iran's curve flattens and dips, signalling a collapse in compounding. By contrast, South Korea, Taiwan, and Singapore enter multi-decadal high-slope regimes during the 1980s–1990s. This fundamental divergence in *rates* explains why Iran's strong 2000s recovery does not translate into convergence.

Similarly, when compared to established systems (Panel B), the Netherlands and Israel trace mature, high-level trajectories with moderately rising slopes, consistent with large, stable research systems. Relative to this group, Iran's post-1979 dip is distinctive: the level falls below Greece in the 1980s and only re-approaches the regional band after 2000. The key contrast is regime stability: the mature systems preserve low implied doubling times throughout the 1980s–1990s, whereas Iran shifts into a high-doubling-time regime.

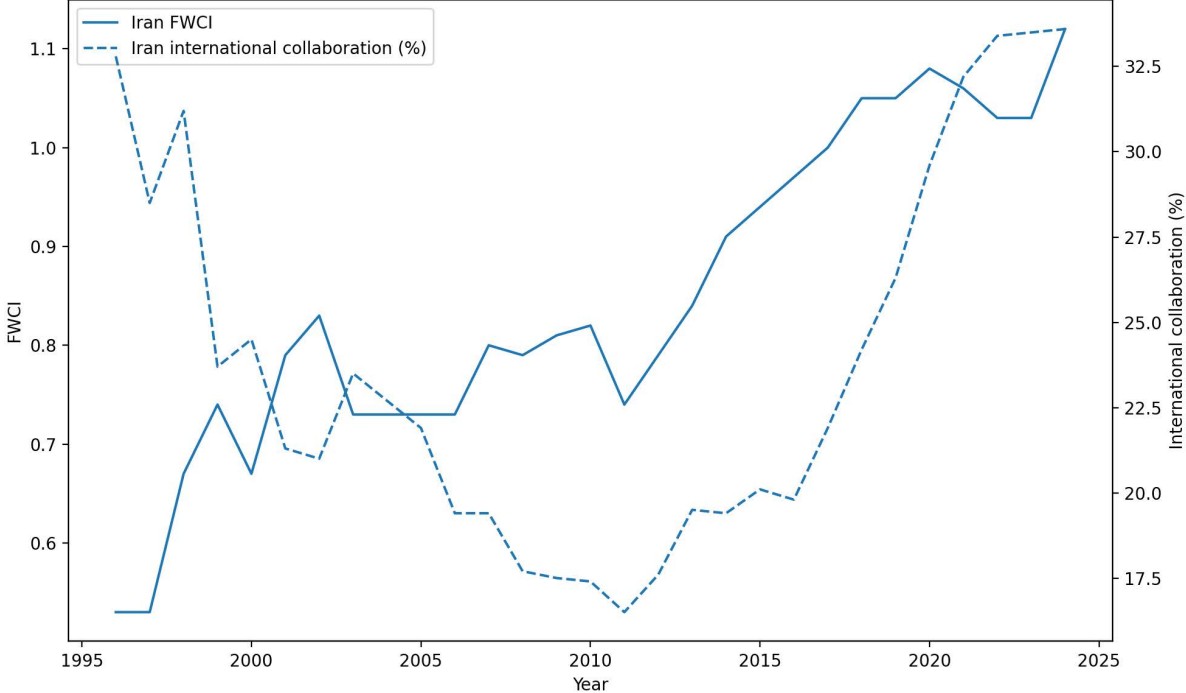

**Fig 14. Iran's observed FWCI and international collaboration share (1996–2024).** The figure shows that changes in Iran's indexed citation impact occurred alongside substantial variation in international collaboration intensity. This does not establish causality by itself, but it supports a more cautious interpretation in which observed FWCI reflects not only domestic incentive structures, but also changing levels of international integration and visibility.

**Table 1. Comparative Scientometric Profile of Iran and Selected Comparator Systems. Total output values are reported for 1996–2024, FWCI values are averaged over 2015–2024, and Top 10% values are averaged over 2015–2023.**

| Country | Total Output | Avg. FWCI | Avg. Top 10% |
|---|---|---|---|
| | (1996–2024) | (2015–2024) | (2015–2023) |
| Iran | 980,305 | 1.033 | 12.8 |
| South Korea | 1,736,423 | 1.110 | 13.3 |
| Taiwan | 935,554 | 1.132 | 11.9 |
| Singapore | 472,133 | 1.891 | 22.7 |
| Netherlands | 1,443,404 | 1.770 | 19.4 |
| Israel | 549,328 | 1.483 | 14.0 |

Finally, the comparison with China (Panel C) illustrates a bifurcation of paths. While Iran experiences a trough and slow rebuild, China undergoes a two-stage regime shift: a steepening in the 1990s and an even steeper rise post-2000. The persistent slope gap through the 1990s–2010s makes absolute catch-up arithmetically implausible for Iran, even with its robust 2000s growth.

### 4.7. Per-capita capacity

This population-normalized view also qualifies the late-1970s level comparison discussed earlier: while Iran's absolute output in 1978 exceeded South Korea and was broadly similar to Taiwan, Taiwan's smaller population implies stronger per-capita publication intensity at that time. Normalizing by population (Fig 16) clarifies that the Asian Tigers exhibit early

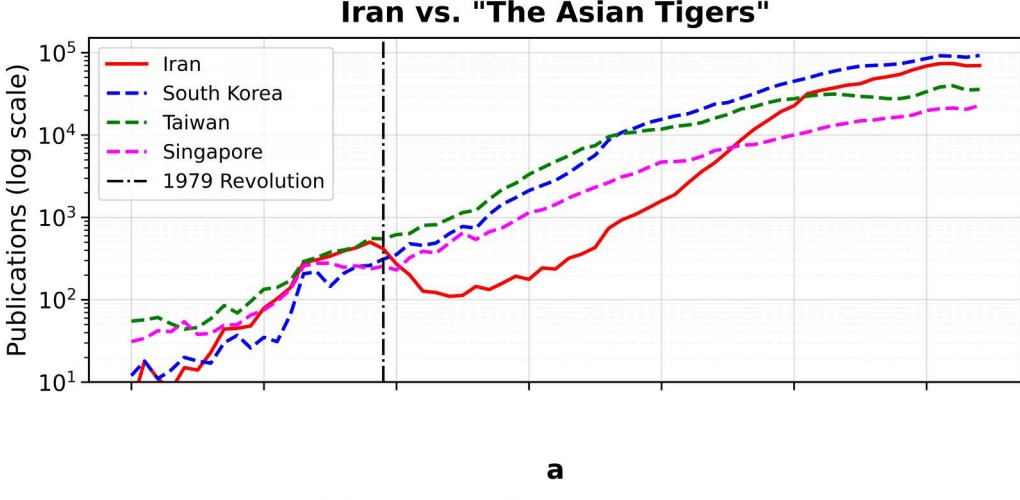

a

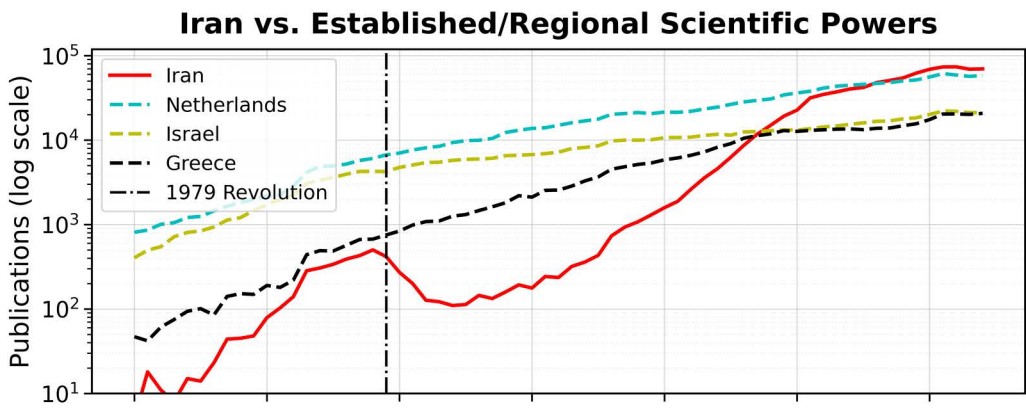

b

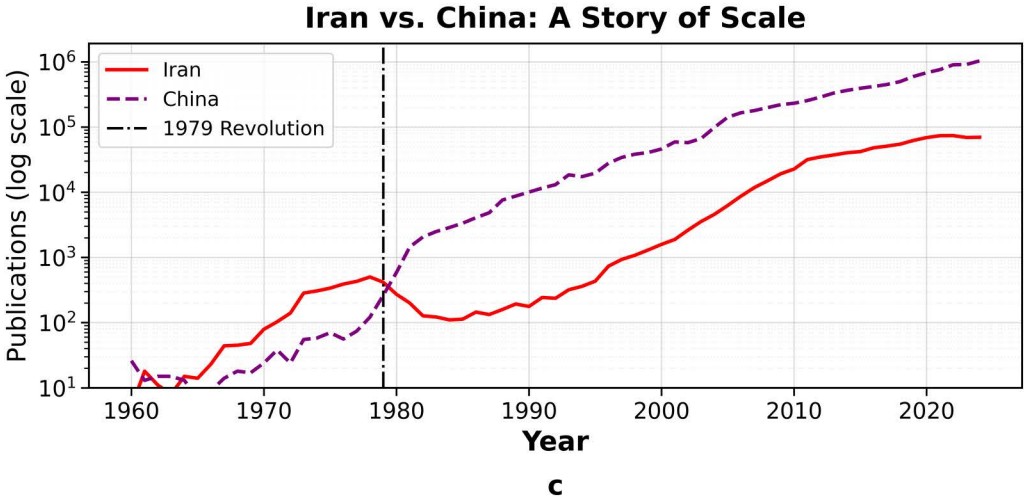

c

**Fig 15. Comparative growth trajectories for Iran and its peer groups.**

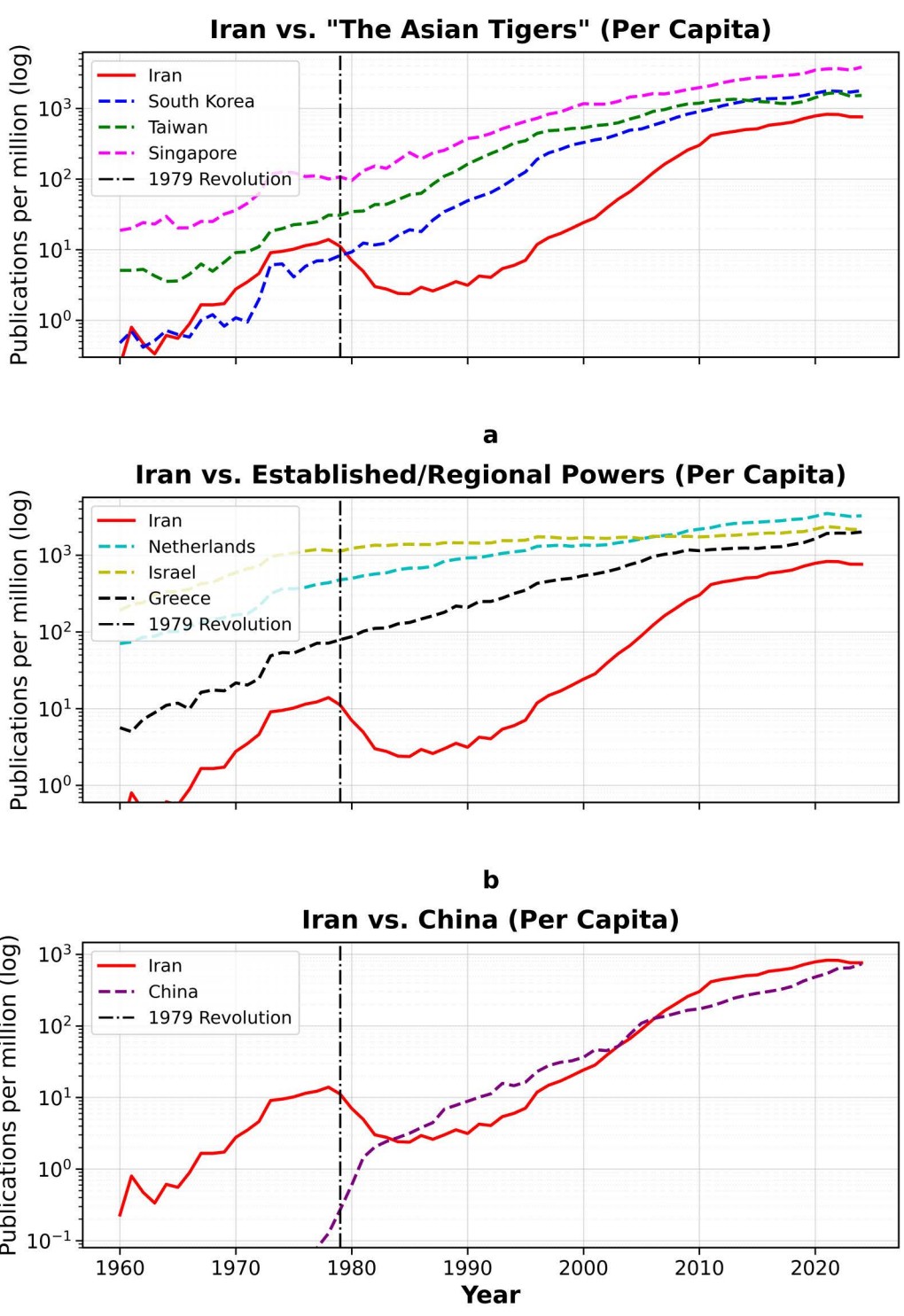

**Fig 16. Publications per million people (log scale).** Normalization by population reveals capacity per person. Iran's post-1979 dip and delayed catch-up contrast with the continuous climb of the Asian Tigers and mature systems.

inflection and multi-decadal compounding on a per-capita basis, reflecting institutional deepening (graduate training, research careers, internationalization). Iran's post-1979 trough is stark on this metric: per-capita output falls sharply, then climbs in the 2000s, but the *relative* gap with Tigers and mature systems (Netherlands, Israel) remains large. The normalization shows Iran's 2000s recovery was driven primarily by scale rebuilding rather than a per-capita surge sufficient for convergence.

### 4.8. Scientific growth momentum: Implied years to double output

Fig 17 tracks *implied years to double* publication output—a momentum metric computed from rolling growth rates (lower values indicate stronger compounding). The vertical dashed line marks 1979.

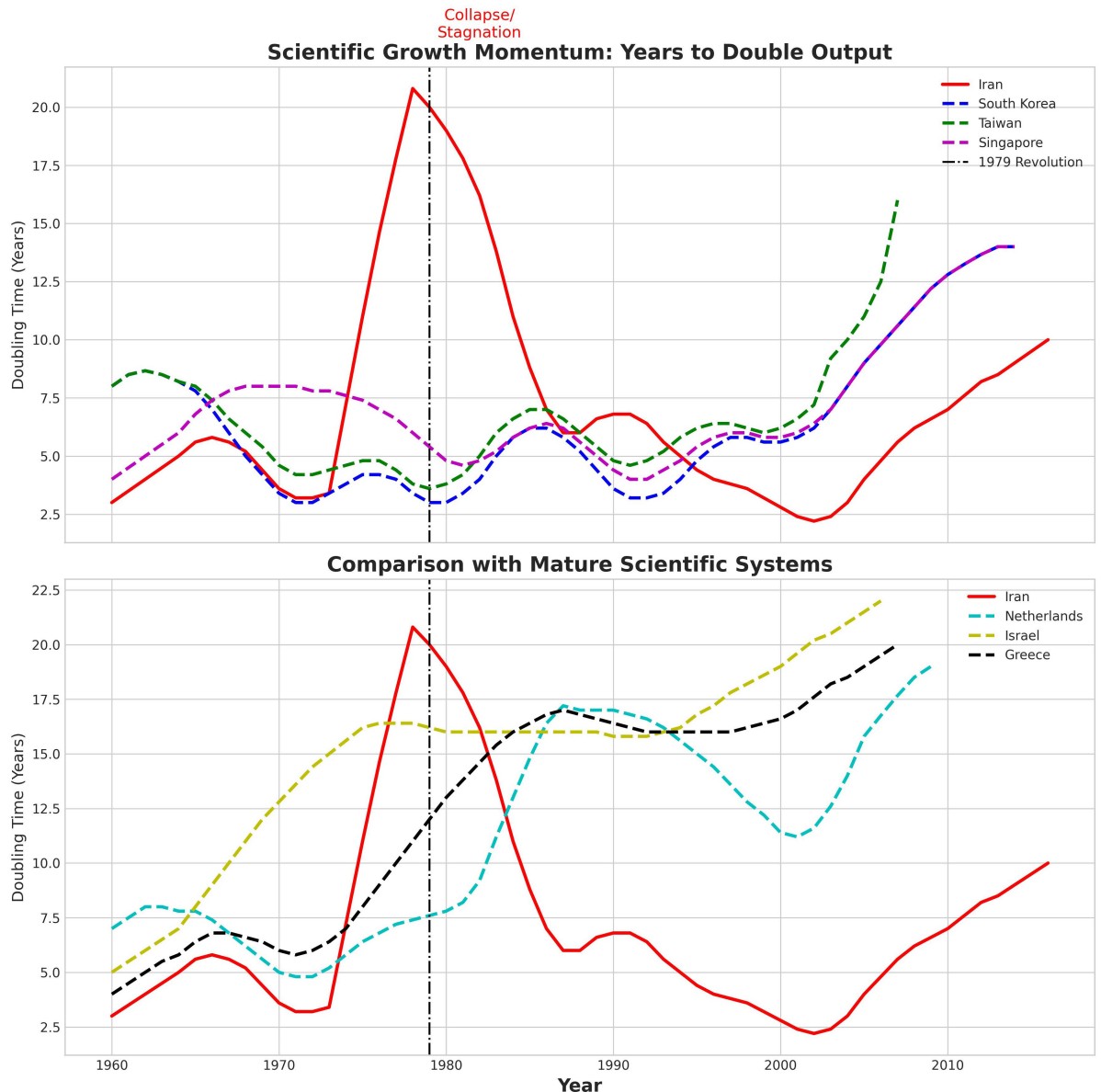

**Fig 17. Growth momentum and implied doubling times.**

**4.8.1. Panel A (top): Iran vs. the Asian Tigers.** Iran enters the 1970s with comparatively *low* doubling times (about 3–5 years), signalling an active, fast-growing system. Around 1979–1981 the red curve spikes abruptly to well above 20 years, a signature of stagnation; this regime shift coincides with the revolution and its immediate sequelae. Momentum then gradually normalizes in the late 1980s (back toward ~6–7 years), briefly improves around 2000 (approaching ~3 years), and subsequently deteriorates again during the 2010s (rising toward ~8–10 years).

In sharp contrast, South Korea and Taiwan maintain *persistently low* doubling times through the 1980s–1990s (roughly ~3–6 years), exactly the regime required for multi-decadal exponential scale-up. Singapore shows a similar pattern: momentum strengthens through the 1980s–1990s, then lengthens in the 2010s as a maturing system naturally slows. Panel A shows that the Tigers *sustain* low doubling times for long horizons just as Iran transitions into a high-doubling-time regime circa 1979; even Iran's later recovery does not keep momentum low for long enough to deliver convergence.

**4.8.2. Panel B (bottom): Iran vs. mature/regional scientific systems.** The Netherlands and Israel display the characteristic profile of *mature* research ecosystems: doubling times hover in the low-teens and gradually lengthen over time, reflecting steady compounding from a high base rather than explosive growth from a low base. Greece climbs from mid-single-digit doubling times in the 1960s toward the teens by the 1990s and 2000s, consistent with long-horizon capacity building in a smaller system. Against this backdrop, Iran's trajectory is distinctive: a pre-1979 low-doubling-time regime (fast momentum), a dramatic spike to >20 years at the turning point, a late-1980s normalization, a short early-2000s acceleration, and then a renewed slowdown.

## 4.9. Comparison-based and scenario-based counterfactual benchmarks

Throughout the counterfactual discussion below, annual divergences are reported as *annual gaps*, whereas long-run accumulated losses are reported as *cumulative shortfalls* or, equivalently in the manuscript's shorthand, the *knowledge deficit*.

Fig 18 illustrates the logic of comparing Iran's observed trajectory with both a comparison-based benchmark and a scenario-based developmental-growth benchmark. The purpose of this comparison is not to identify a single definitive alternate history, but to show how different benchmark constructions frame the magnitude of the long-run shortfall.

The SCM trajectory shown here is generated from a weighted donor combination selected on the basis of pre-treatment fit. As reported below, the baseline SCM is heavily influenced by China and should therefore be interpreted primarily as an optimization-based comparison benchmark rather than as a historically literal analogue. By contrast, the scaled-Korea path is not part of the SCM procedure; it is a developmental-growth proxy scenario that applies South Korea's post-1979 growth dynamics to Iran's late-1970s base. For this reason, the scaled-Korea series is best understood as an upper-range illustrative scenario rather than as a central estimate.

Read in this way, Fig 18 makes two points. First, Iran's post-1979 trajectory diverges sharply from both benchmark types after the interruption period. Second, the scale of that divergence depends strongly on benchmark construction. This is precisely why the paper does not rely on any single counterfactual estimate, but instead evaluates the persistence of the break across multiple comparison-based and scenario-based exercises.

## 4.10. Placebo and robustness evidence

A key concern in Synthetic Control Method applications is whether the estimated post-treatment divergence is unusually large relative to placebo reassignments within the donor pool. To address this issue, we conducted in-space placebo tests by iteratively reassigning the 1979 treatment year to each donor country and reconstructing a corresponding synthetic control for that placebo unit.

Fig 19 shows that Iran's post-1979 gap is among the largest in the donor pool. Although it is not the single most extreme placebo in every comparison, it clearly belongs to the upper tail of post-treatment deviations. This result strengthens the interpretation that the observed divergence is not simply an artifact of pre-treatment fit, but reflects a substantial structural break associated with the 1979 disruption.

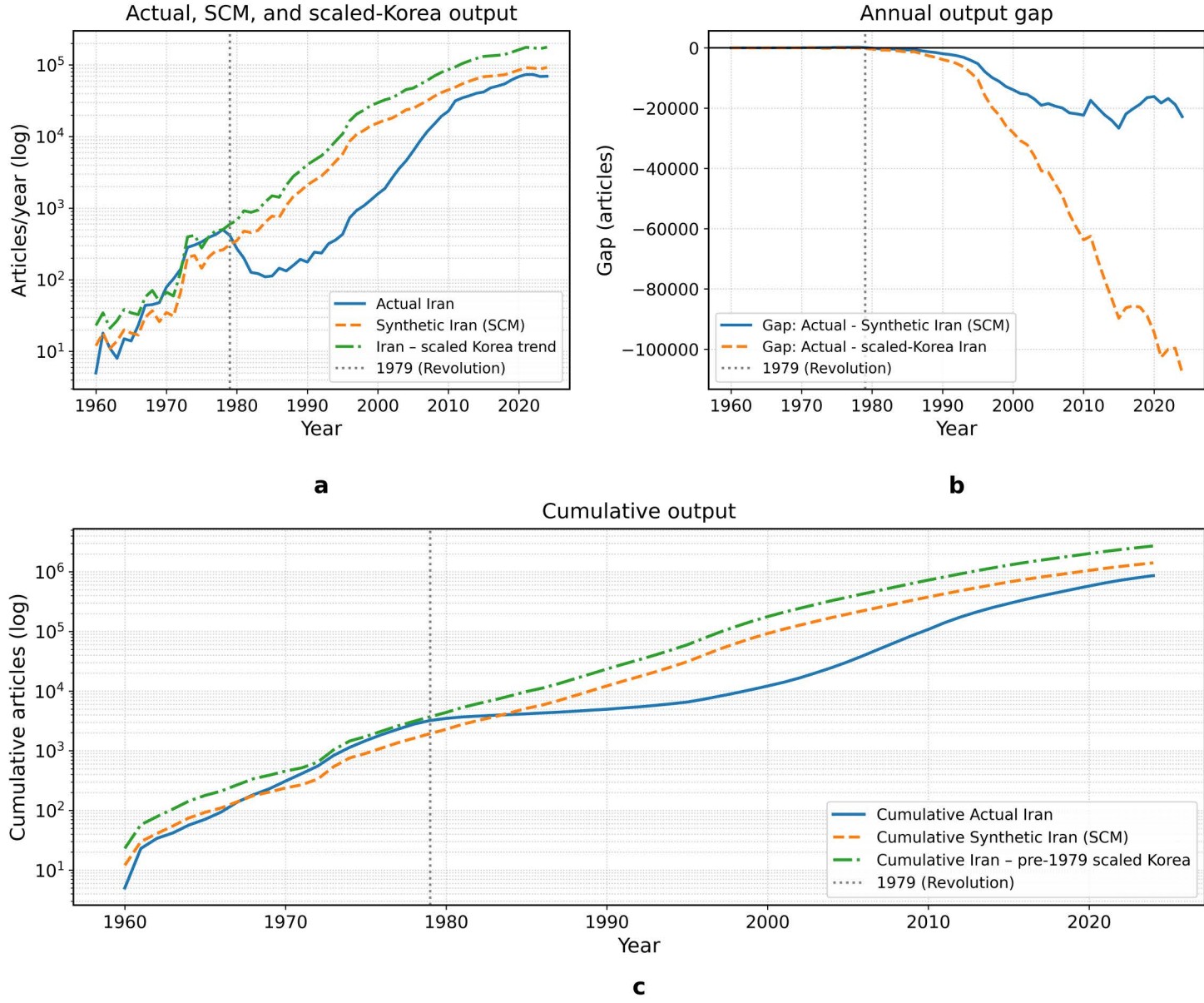

**Fig 18. Comparative evaluation of Iran's scientific trajectory under the SCM benchmark and a separate scaled-Korea growth-proxy benchmark.**

The RMSPE-based robustness checks support the same conclusion. In the baseline SCM specification, Iran's post/pre-treatment RMSPE ratio is very large, and this finding remains stable across alternative predictor specifications. Thus, the qualitative conclusion does not depend on one fragile donor combination or one exact predictor set.

Table 2 reports the donor weights in the baseline SCM specification. The resulting synthetic benchmark is dominated by China, with smaller contributions from Israel, Greece, and Singapore, while South Korea does not receive a positive weight in the multilateral donor construction.

To clarify why China receives the dominant donor weight in the SCM specification, it is important to note that SCM does not select donors on the basis of historical narrative similarity, contemporary geopolitical status, or present-day scientific scale

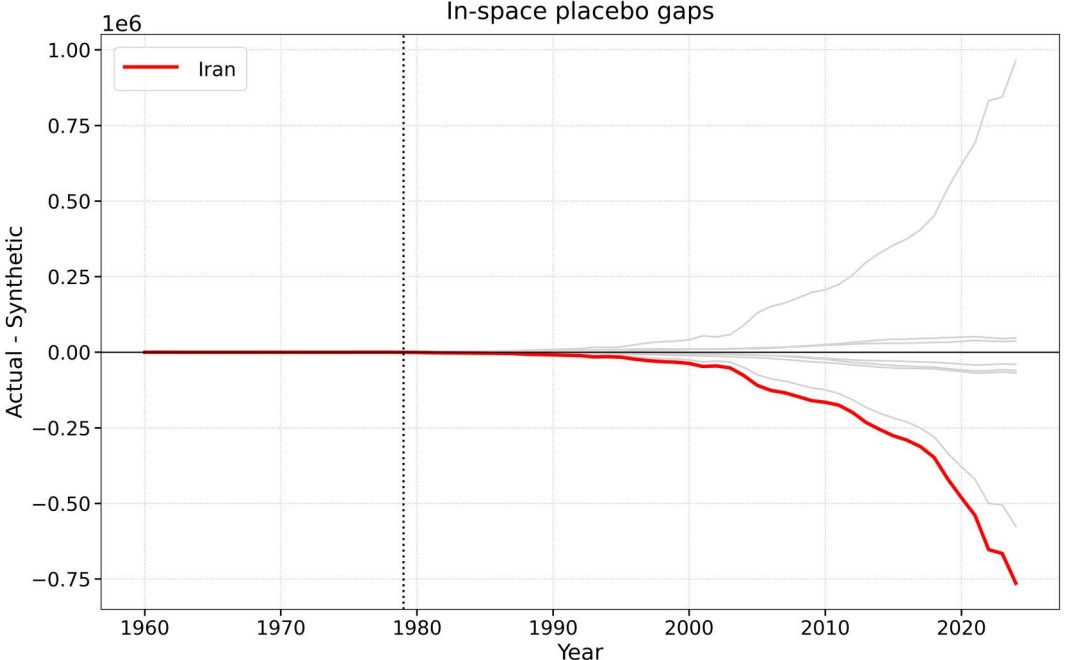

**Fig 19. In-space placebo gaps for the SCM specification.** The red line corresponds to Iran, while the gray lines show placebo gaps for donor countries. The vertical dashed line marks the 1979 intervention.

**Table 2. Donor weights in the baseline SCM specification.**

| Donor country | Weight |
|---|---|
| China | 0.8028 |
| Israel | 0.0904 |
| Greece | 0.0535 |
| Singapore | 0.0532 |
| South Korea | 0.0000 |
| Netherlands | 0.0000 |
| Turkey | 0.0000 |

alone. Rather, it selects the convex combination of countries that minimizes pre-treatment mismatch in the space of observed predictors and pre-1979 publication trajectories. In our case, China provides the closest numerical contribution to reproducing Iran's pre-1979 profile once GDP per capita, tertiary enrollment, government expenditure, trade openness, and the lagged publication path are considered jointly. Smaller weights on Israel, Greece, and Singapore further reduce the remaining imbalance.

This result should therefore be interpreted as an optimization outcome rather than as a claim that China is the single most historically analogous case to Iran. More broadly, China should not be read as a theoretically equivalent recovery benchmark for Iran. Its post-1978 scientific expansion occurred under structural conditions – especially global market integration, large-scale state R&D mobilization, and the absence of prolonged sanctions – that differ fundamentally from Iran's post-revolutionary environment. The role of China in the donor pool is methodological: it helps the synthetic control best reproduce the observed pre-1979 characteristics of Iran. For this reason, the SCM benchmark and the separate scaled-Korea benchmark play different roles in the analysis. The former is a comparison-based counterfactual chosen by pre-treatment fit, whereas the latter remains a historically motivated growth-proxy benchmark.

This conclusion is reinforced by specification sensitivity tests. Across alternative predictor sets that exclude trade open-ness, exclude government expenditure, or retain only economic predictors, the post/pre-treatment RMSPE ratio remains very large. The substantive conclusion therefore does not depend on one exact predictor specification.

At the same time, the benchmark-specific magnitude of the inferred shortfall changes materially when China is excluded from the donor pool. In China-excluded and restricted-donor specifications, the qualitative conclusion of post-1979 divergence remains intact, but the implied 2024 and cumulative gaps are substantially smaller than in the China-dominated baseline SCM. This result is important for interpretation: it indicates that the existence of a long-run break is not driven by China alone, even though the quantitative scale of the baseline SCM gap is strongly affected by China's donor weight. For this reason, the China-excluded and restricted-donor SCM results are more informative for the paper's main substantive claim than the baseline SCM point estimates alone. They show that Iran's post-1979 trajectory remains qualitatively below plausible comparison-based benchmarks even when the donor construction is made more conservative and historically restrained. In the revised interpretation adopted here, the baseline SCM is best treated as one benchmark family among several, not as the paper's sole quantitative anchor.

## 4.11. Robustness to an expanded donor pool

As a secondary robustness exercise, we re-estimated the SCM using an expanded donor pool that added Malaysia, Thailand, and Mexico to the original comparison set (Spain was excluded from the expanded donor-pool robustness exercise because its pre-1979 predictor coverage was insufficient for a balanced SCM implementation.). The purpose of this exercise is not to redefine the paper's preferred benchmark, but to assess whether the baseline SCM pattern is merely an artifact of a narrowly restricted donor set.

The expanded-donor specification reproduces the same qualitative pattern as the baseline SCM: Iran tracks the synthetic benchmark reasonably well before 1979 and then falls below it thereafter. However, the expanded solution remains strongly concentrated on China, with smaller positive weights on Israel, Greece, and Singapore. This means that the expanded-donor exercise supports the persistence of post-1979 divergence, but it does not resolve the benchmark-dependence of the SCM's quantitative magnitude. For that reason, the expanded-donor results are best interpreted as confirmatory robustness evidence rather than as the paper's preferred quantitative benchmark.

The robustness diagnostics also support this interpretation. In the expanded-donor specification, Iran's post/pre-treatment RMSPE ratio remains extremely large, and its post-1979 gap continues to lie in the upper tail of placebo reas-signments. Moreover, leave-one-out tests show that the substantive conclusion is not driven by the inclusion or exclusion of any single newly added donor candidate. Taken together, these results strengthen the credibility of the SCM analysis by showing that the central finding is robust not only to predictor choice but also to a materially broader donor pool.

Figs 20–23 present the expanded donor-pool robustness analysis. The results confirm that the main SCM finding is stable: the post-1979 divergence remains large in annual, cumulative, and placebo-gap terms, even after adding Malaysia, Thailand, and Mexico to the donor pool.

Table 3 shows that the expanded donor-pool solution remains concentrated on China, Israel, Greece, and Singapore, while the newly added donor candidates do not receive positive baseline weights. This indicates that broadening the donor pool does not materially alter the optimized synthetic benchmark.

## 4.12. Benchmark sensitivity across developmental-growth proxy families

The developmental-growth proxy exercises reinforce the same general conclusion while also showing clearly that the cumulative magnitude of the inferred shortfall depends on benchmark choice. When South Korea's post-1979 growth dynamics are used as a proxy, the implied cumulative gap is very large, placing this benchmark in the upper range of the scenario set. By contrast, Taiwan-based and Turkey-based proxy constructions yield much smaller implied gaps and, in some specifications, only modest divergence by the terminal years. Proxy families based on multi-country developmental averages occupy an intermediate position between these extremes.

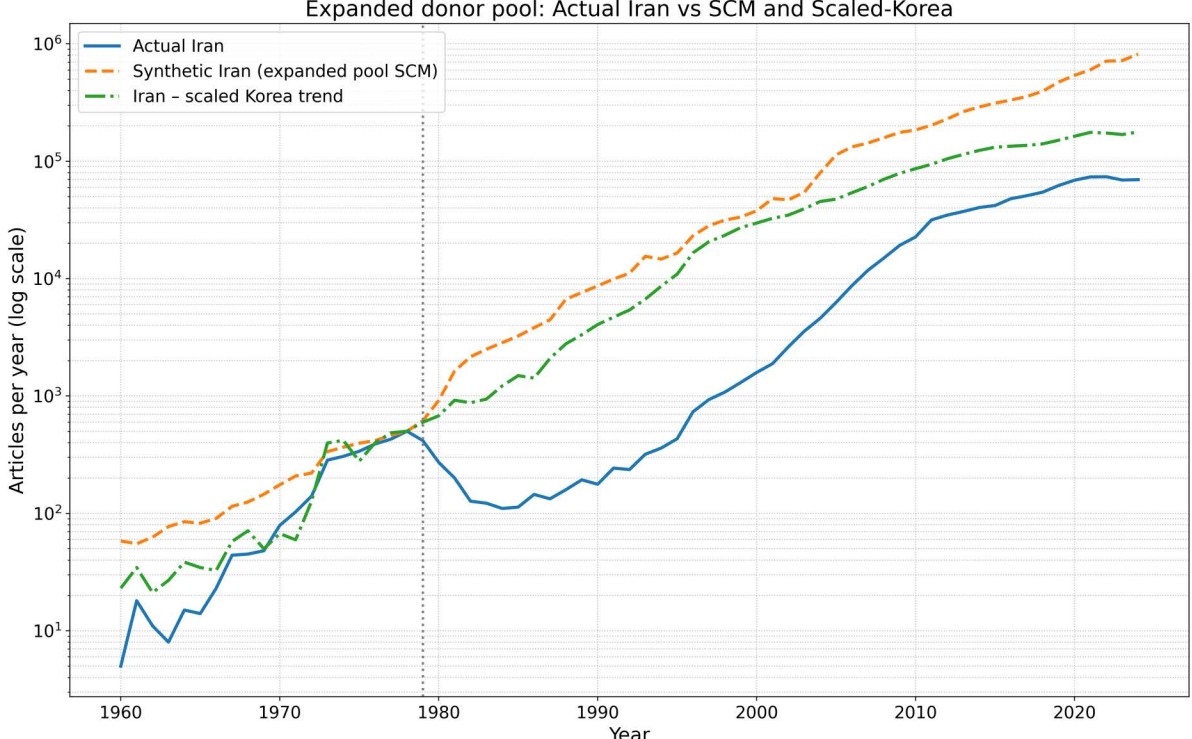

**Fig 20. Robustness check using an expanded donor pool.** Actual Iran is compared with the synthetic control generated from the expanded donor set and with the separate scaled-Korea benchmark. The qualitative post-1979 divergence remains unchanged.

This pattern has two implications. First, it confirms that the article should not rely on South Korea alone as the central quantitative counterfactual. Second, it shows that the paper's core contribution lies less in defending one exact cumulative publication-loss figure than in demonstrating that the post-1979 interruption permanently altered Iran's compounding scientific trajectory. In the revised interpretation adopted here, South Korea is therefore retained as an illustrative upper-range developmental benchmark, while more moderate multi-country proxy families provide a more balanced picture of benchmark sensitivity.

## 5. Mechanisms of divergence: A comparative policy analysis

The scientometric evidence indicates that a major divergence emerged after 1979; this section turns from description to interpretation. Specifically, it examines how differences in institutional continuity, incentive structures, and state–science relations may have contributed to the widening gap between Iran and a set of developmental and high-performing comparator systems.

### 5.1. Legal frameworks and institutional stability

Developmental-state cases such as South Korea illustrate how scientific expansion can be tied to industrial strategy through relatively stable institutional frameworks. In South Korea, the enactment of the *Technology Development Promotion Law (1972)* was a watershed moment, providing fiscal incentives for private firms to internalize R&D and mandating the adoption of indigenous technologies [17]. This was supported by the creation of the *Economic Planning Board* (EPB), a pilot agency with the power to align science budgets with national five-year plans.

By comparison, Iran's pre-revolutionary "Great Civilization" push did not develop the same kind of binding legislative infrastructure. While Iran established the *Plan and Budget Organization* (PBO), it functioned more as an allocator of oil

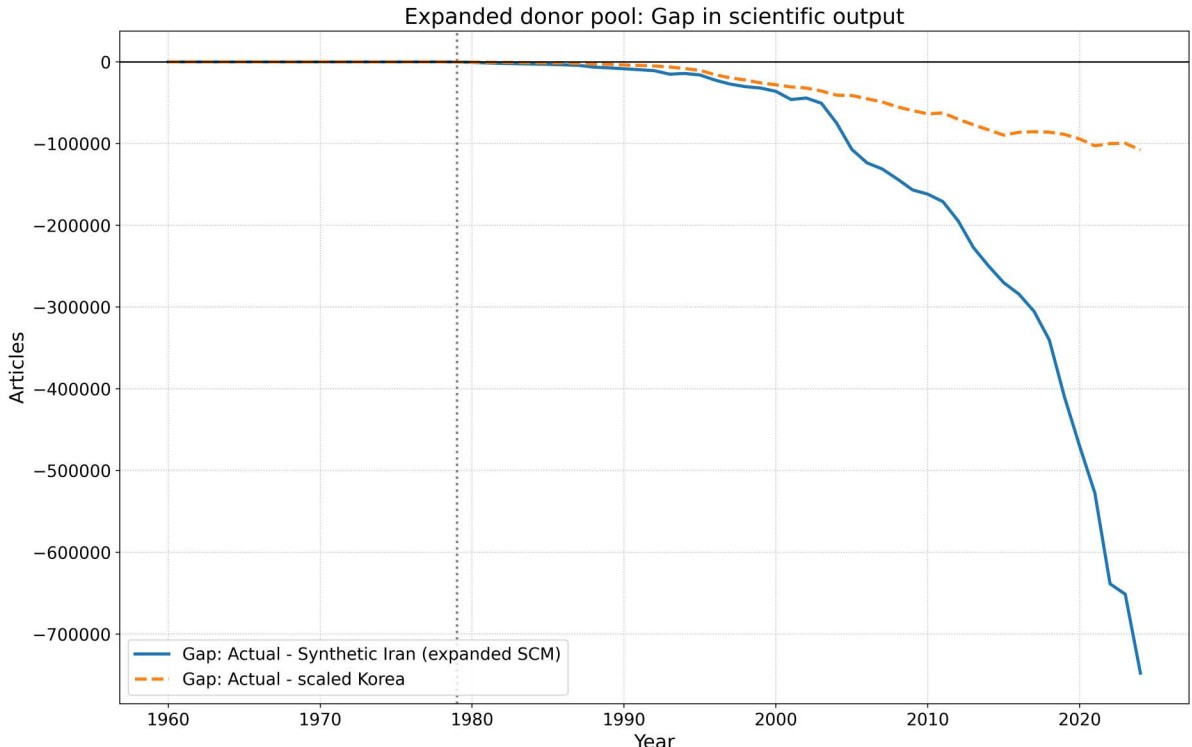

**Fig 21. Annual output gap under the expanded donor-pool robustness analysis.** Iran's observed trajectory remains substantially below both the expanded SCM benchmark and the scaled-Korea benchmark after 1979.

rents than a disciplinarian of industrial performance. Following the 1979 Revolution, the "Cultural Revolution" (1980–1983) effectively dismantled the existing governance structures. The dissolution of university autonomy and the centralization of faculty hiring under the *Supreme Council of the Cultural Revolution* severed the continuity of research lineages that South Korea was meticulously cultivating at institutions like KAIST (Korea Advanced Institute of Science and Technology).

### 5.2. The Quantity-Quality paradox and incentive structures

The post-2000 recovery in Iran's publication volume (Fig 9), without a commensurate rise in Field-Weighted Citation Impact (FWCI) (Fig 10), can be interpreted partly in light of specific incentive structures introduced by the Ministry of Science, Research, and Technology (MSRT). To restore scientific standing, the state tied faculty promotion and doctoral graduation requirements directly to the number of indexed papers [18].

   In this paper, "export discipline" is not used to imply that Iran should have replicated South Korea's export structure literally. Rather, the term refers more broadly to a policy environment in which scientific and technological investment is disciplined by external performance criteria, such as competitiveness, industrial upgrading, technological absorption, and demand from productive sectors. In the South Korean case, these disciplining pressures were closely tied to manufactured-export performance. In Iran's case, an analogous disciplining mechanism would have meant stronger links between science funding and non-rentier productive outcomes—for example, technological capability in tradable sectors, industry-oriented research demand, innovation uptake, and performance-based evaluation beyond publication counts alone. The argument, therefore, is not that Iran lacked exports in a narrow accounting sense, but that its scientific system was less effectively anchored to productivity-enhancing feedback from competitive productive sectors.

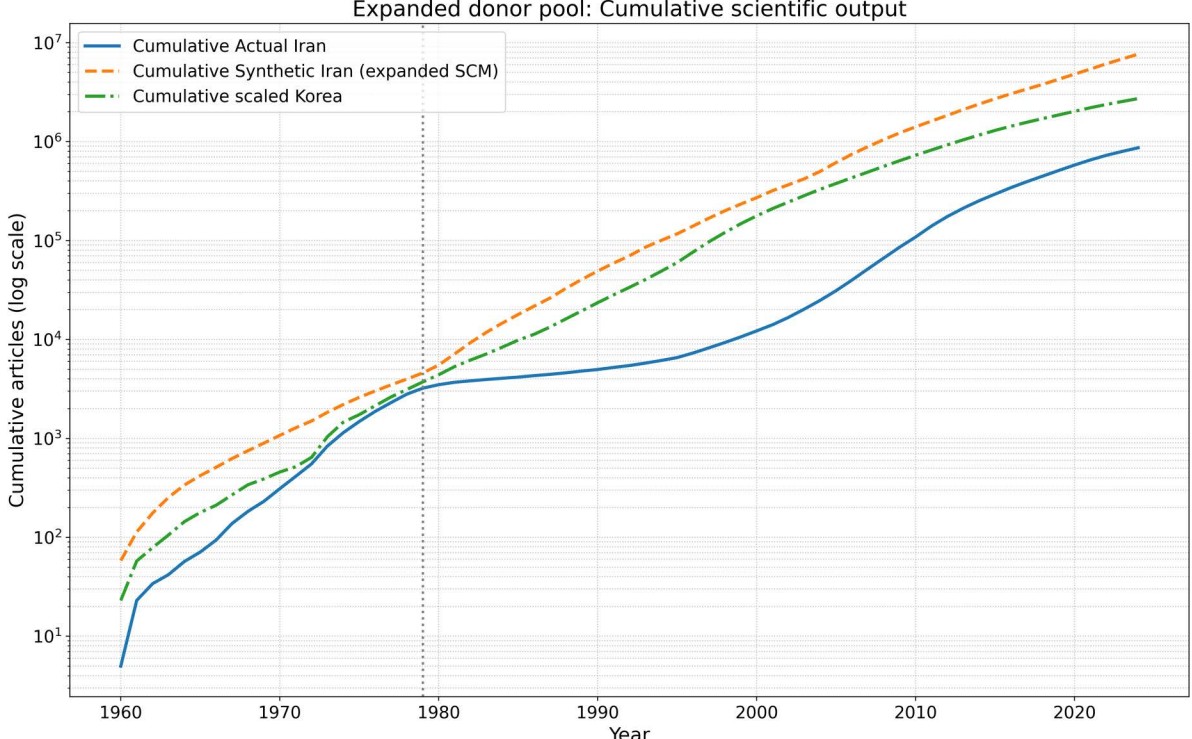

**Fig 22. Cumulative scientific output under the expanded donor-pool robustness analysis.** The long-run cumulative gap remains large, confirming that the post-1979 interruption produced a durable loss of compounding capacity.

This policy created a "Goodhart's Law" effect: by making publication counts the target, the metric ceased to be a valid measure of scientific capacity [19]. In South Korea, conversely, the 1990s saw a shift from government-led R&D to private-sector-led R&D (e.g., Samsung, LG), where the metric of success was not publication volume but patent viability and global market share. This contrast also helps explain why developmental-growth proxy scenarios based on South Korea generate much larger implied shortfalls than more moderate benchmark families: they reflect a science system more tightly linked to industrial demand, external performance discipline, and competitive productive-sector feedback, whereas Iran's post-revolutionary trajectory was shaped more strongly by bureaucratic, rent-conditioned, and publication-centered administrative incentives.

The rentier interpretation also requires a temporal qualification in the post-2000 period. The manuscript does not assume that Iran's oil economy remained unchanged after 2000; on the contrary, the later period was marked by sanctions-related revenue compression, greater macro-fiscal instability, and repeated pressure for diversification. These changes matter because they imply that the rentier mechanism should be understood less as a simple story of abundant oil rents automatically financing science, and more as a broader institutional pattern in which research funding remains vulnerable to externally driven fiscal cycles, administrative reprioritization, and weak industrial embedding. In that sense, reduced oil rents do not necessarily invalidate the rentier-state interpretation; they may instead expose its institutional fragility more clearly. Table 4 summarizes the contrast between the developmental-state logic exemplified by South Korea and the more centralized, rent-conditioned, and publication-centered policy environment that shaped Iran's post-revolutionary scientific trajectory.

## 6. Policy Implications: Escaping the trap

For Iran, responding to the long-run scientific shortfall documented in this study requires more than simply expanding publication output. The central challenge is to reorient the scientific system toward impact, continuity, and long-run efficiency

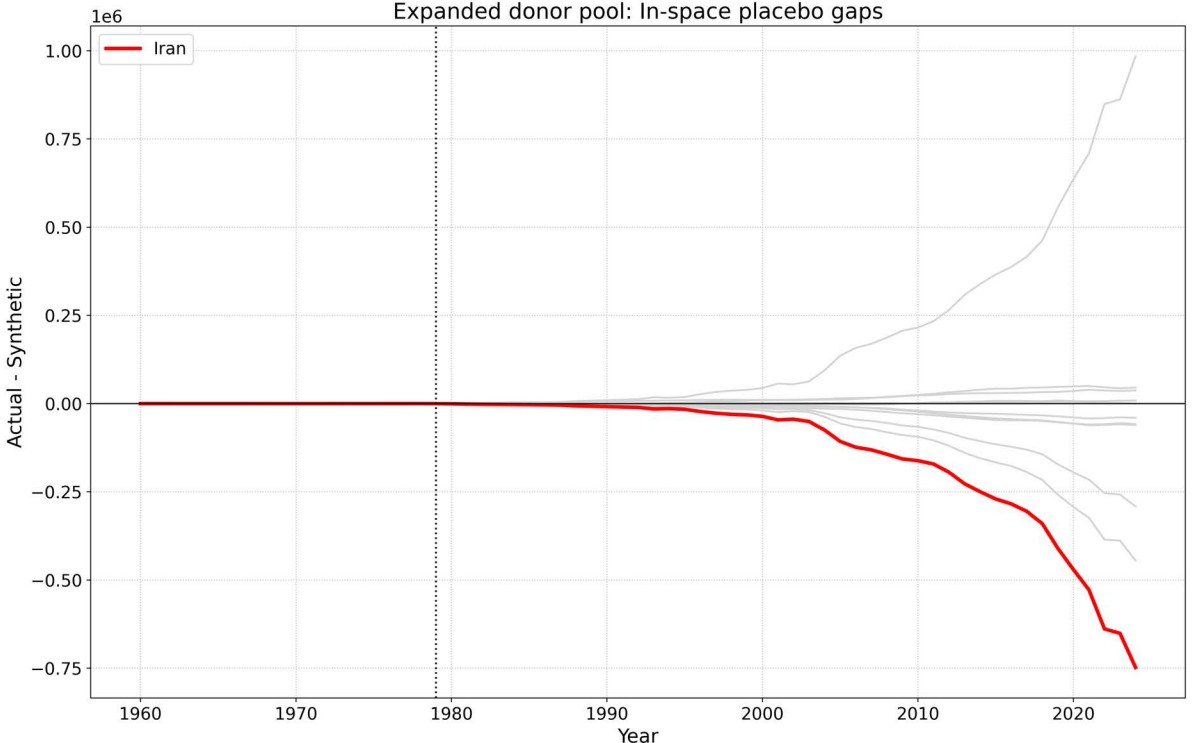

**Fig 23. In-space placebo gaps for the expanded donor-pool SCM specification.** The red line corresponds to Iran, while the gray lines show placebo gaps for donor and pseudo-treated countries. Iran remains in the upper tail of post-treatment deviations.

**Table 3. Donor weights in the expanded donor-pool SCM specification.**

| Donor country | Weight |
| --- | --- |
| China | 0.7868 |
| Israel | 0.0802 |
| Greece | 0.0710 |
| Singapore | 0.0619 |
| South Korea | 0.0000 |
| Turkey | 0.0000 |
| Netherlands | 0.0000 |
| Malaysia | 0.0000 |
| Thailand | 0.0000 |
| Mexico | 0.0000 |

rather than volume alone. Drawing from developmental-state experience, broader innovation-systems scholarship, and the Middle-Income Trap literature, we propose three policy pivots:

1. **From Volume to Impact:** The MSRT must reform faculty promotion criteria to deprioritize raw publication counts. Evaluation systems should weight citation impact (FWCI), international collaboration, and industrial relevance higher than the sheer number of articles. This is necessary to curb the fragmentation of research into minimally differentiated publishable units (often described as "salami slicing").

**Table 4. Comparative Policy Divergence (1980–2000).**

| South Korea (Developmental State) | Iran (Post-Revolutionary/Rentier) |
|---|---|
| **Mechanism:** Science policy disciplined by export performance, industrial upgrading, and competitive productive-sector feedback. | **Mechanism:** Science policy shaped more by state allocation, ideological control, and publication-based administrative incentives than by competitive productive-sector feedback. |
| **Funding:** Tied to export performance and industrial upgrading. | **Funding:** Strongly conditioned by rent-dependent public finance, with volatile and often prestige-oriented allocation logic. |
| **Goal:** Industrial Competitiveness (Quality/Utility). | **Goal:** Ideological Purity & Statistics (Quantity/Volume). |
| **Result:** High FWCI, High Patents. | **Result:** High Volume, Low Impact (MIT). |

2. **Re-integrating the Diaspora:** Some developmental-state systems, including South Korea, partially reversed earlier brain-drain dynamics by creating high-autonomy research institutions and attractive return pathways for internationally trained scholars. Iran possesses a large scientific diaspora, and a credible reintegration strategy would require protected research environments, competitive funding, and institutional arrangements that reduce political and bureaucratic friction for collaboration and return.

3. **Breaking the Rentier Link:** Science funding must be insulated not only from the long-run political logic of rent dependence, but also from the shorter-run fiscal instability associated with oil-price shocks, sanctions-related revenue compression, and cyclical budget stress. A National Science Foundation–style model, funded by a sovereign wealth endowment and administered through competitive peer review, could help create a more counter-cyclical and institutionally stable funding base for research. The goal is therefore not merely to reduce dependence on oil abundance, but to protect scientific investment from the volatility and reprioritization pressures that characterize rent-dependent fiscal systems.

## 6.1. Implementation constraints and institutional feasibility in Iran

These policy recommendations should not be interpreted as technocratic reforms that could be implemented independently of Iran's existing political and administrative structure. In practice, major changes in university governance, promotion criteria, or protected research autonomy would have to pass through a layered institutional environment shaped by the Ministry of Science, Research, and Technology (MSRT), the Plan and Budget Organization (PBO), and, more fundamentally, the Supreme Council of the Cultural Revolution, which has long exercised significant influence over higher education governance, faculty regulation, and the ideological boundaries of university administration. For this reason, the practical policy question is not simply what an optimal reform package would look like, but which reforms are institutionally feasible under conditions of centralized oversight and bureaucratic path dependence.

A realistic interpretation of the policy implications is therefore incremental rather than maximalist. For example, a shift from volume-based to impact-sensitive evaluation would more plausibly begin through partial revision of promotion and doctoral-completion criteria within selected universities or research institutes, rather than through an immediate system-wide transformation. Likewise, any effort to re-engage the scientific diaspora would likely require semi-protected institutional channels—such as internationally collaborative research centers, competitively funded laboratories, or specialized graduate institutes with greater procedural autonomy—rather than assuming a wholesale liberalization of the university system. In this sense, the creation of "islands of excellence" should be understood not as a simple slogan, but as a politically constrained strategy of pilot institutional differentiation within a centralized system.

At the same time, bureaucratic resistance should be expected. Publication-count-based evaluation, centralized approval structures, and administratively distributed research resources create vested interests that may resist reform even when those arrangements are suboptimal for scientific impact. For this reason, politically feasible reform would likely depend on sequencing and coalition-building: beginning with pilot programs, demonstrating measurable gains in international visibility and research quality, and then using those results to justify broader institutional adaptation. The implication

is that the main obstacle is not only resource scarcity, but also the governance structure through which scientific policy is authorized, implemented, and defended.

## 7. Conclusion

This paper provides a comprehensive, data–driven reconstruction of Iran's scientific trajectory over more than six decades and evaluates its long-term evolution through multiple counterfactual benchmark families. The findings demonstrate that the disruption triggered by the 1979 Revolution produced a structural break whose effects extend beyond the immediate decline in scientific activity. Rather than representing a temporary interruption, the post–1979 decade reset the underlying growth rate of Iran's scientific system and altered its long–run development path.

The counterfactual analyses clarify this divergence through multiple benchmark families rather than a single reconstructed path. Comparison-based SCM specifications, donor-pool robustness checks, and developmental-growth proxy scenarios all point to the same qualitative conclusion: Iran's post-1979 scientific trajectory remained durably below plausible counterfactual benchmarks even after the later recovery in publication volume. At the same time, the cumulative magnitude of the implied shortfall varies substantially across benchmark constructions. This is especially evident when comparing China-influenced SCM specifications, more conservative China-excluded donor constructions, and developmental-growth proxy scenarios based on different comparator families.

Taken together, the SCM evidence, donor-pool robustness checks, placebo tests, and benchmark-sensitivity exercises suggest that the post-1979 interruption was not merely a short-run decline in scientific activity. It produced a long-lasting loss of compounding capacity whose effects remained visible decades later. The central implication is therefore not simply that Iran lost output during the years of disruption, but that it lost a growth regime that was never fully reconstituted on the same trajectory.

The comparison between these reconstructed trajectories and Iran's observed performance highlights a key insight: Iran's scientific system successfully regained scale in the post–2000 period, but it did so along a shallower curve than its historical peers. The resulting long-run shortfall is therefore not merely the product of lost years, but the cumulative consequence of a lost growth rate.

At the same time, the paper does not claim that all political disruptions produce the same long-run scientific consequences. Iran's experience reflects a historically specific configuration: revolutionary rupture within a surviving centralized state, prolonged war, ideological restructuring of universities, and later recovery under publication-centered incentives. Other disruptions—such as post-Soviet state dissolution—involve different institutional pathways and should not be treated as directly equivalent. The contribution of this paper is therefore not to universalize Iran's experience, but to show how one particular form of political rupture generated a durable break in scientific compounding.

In sum, the evidence demonstrates that Iran's scientific development has been shaped not only by its capacity to produce publications but by the structural conditions that enable sustained, compounding growth. The counterfactual benchmarks presented in this study illuminate the generational cost of the disruption that occurred in 1979 while also showing that the quantitative scale of that cost depends on benchmark choice. The broader conclusion is nonetheless clear: institutional continuity, openness, and incentive structures matter profoundly for long-run scientific compounding. Reversing the shortfall documented here therefore depends less on restoring publication volume alone than on rebuilding the deeper conditions that sustain durable scientific momentum. The robustness and benchmark-sensitivity analyses reported in Appendix A further support the conclusion that the qualitative evidence for post-1979 divergence is stable, even though the quantitative magnitude of the cumulative shortfall remains benchmark-dependent.

## Appendix A. A Counterfactual modeling and robustness evidence

### A.1 Full empirical results

To assess the long-run magnitude of Iran's post-1979 divergence, we compare the observed publication trajectory with several counterfactual benchmark families rather than relying on a single reconstructed path. This strategy reflects the main methodological conclusion of the paper: the qualitative finding of a durable post-1979 break is robust across approaches, whereas the quantitative magnitude of the cumulative shortfall is sensitive to benchmark choice.

Accordingly, the counterfactual exercises reported below should be interpreted as benchmark-specific estimates rather than as interchangeable measurements of one fixed quantity. Where cumulative publication gaps are summarized in rounded form, small numerical differences may arise from specification choice, year-end conventions, or presentation rounding, without affecting the paper's qualitative conclusion.

### A.1.1 Model B (Perpetual Exponential Growth)

This model serves as a theoretical benchmark to illustrate the power of compounding. It assumes that the institutional momentum Iran had built up in the decade prior to the revolution would continue indefinitely. We calculate the Compound Annual Growth Rate (CAGR) from 1970 to 1978, which stood at a robust $r \approx 13.1\%$. The projection is then calculated using the standard exponential growth formula:

$$\hat{N}^B_{IR,t} = N_{IR,1978} \times (1 + r)^{(t-1978)}$$

As shown by the dotted gray line in Fig 24, this model projects an unrealistic output of over 1.2 million papers by 2024. While no system can sustain such a high growth rate forever, it highlights a significant mathematical limitation for Iran's lost potential, suggesting a trajectory that could have theoretically rivaled the scale of modern-day China.

### A.1.2 Model C (Logistic S-Curve Growth with Carrying Capacity)

This model introduces a significant element of realism: the concept of maturation. Scientific systems, like all growth processes, eventually face diminishing returns as they exhaust available resources (e.g., funding, human capital, institutional capacity), causing their growth to slow and follow an S-shaped curve. The logistic model captures this by incorporating a "carrying capacity" ($K$), which represents the plausible upper limit of a nation's scientific system.

**Justification for Carrying Capacity (K):** We set $K = 150{,}000$ publications per year. This figure is not arbitrary; it is chosen to represent the scale of a mature, top-tier, and stable scientific nation, comparable to the current output of countries like South Korea, Italy, or Canada. In the context of scientometrics, K is a function of sustained investment, population size, and effective science policy. Nations achieving this scale of output typically maintain a Gross Expenditure on R&D (GERD) consistently above 2% of GDP, coupled with a high density of researchers within the population. The value of 150,000 thus represents a plausible limitation that a nation with Iran's demographic potential and pre-1979 momentum could have realistically achieved, had it maintained stability and adopted similar long-term investment strategies.

The logistic formula is:

$$\hat{N}^C_{IR,t} = \frac{K}{1 + \left( \frac{K}{N_{1978}} - 1 \right) e^{-r(t-1978)}}$$

The dashed blue line in Fig 24 shows this path. Iran's growth is initially rapid, but then subsequently slows as it approaches the 150,000-publication mark. This model projects a far more plausible scenario where Iran would have become a major, stable scientific power.

### A.1.3 Model D (South Korea as a Growth-Proxy)

This final model is compelling as it is grounded in a real-world success story. It answers the question: "What if Iran, starting from its superior position in 1978, had simply followed South Korea's subsequent path of development?"

**Justification for Proxy Selection:** The choice of South Korea as a proxy is theoretically motivated. In comparative research, suitable proxies are those that share critical baseline characteristics but differ on the key variable of interest (in this case, political stability post-1979). Prior to 1979, both Iran and South Korea were on a similar trajectory of state-led modernization and industrialization, with science and technology policy identified as a central pillar of national development. South Korea's

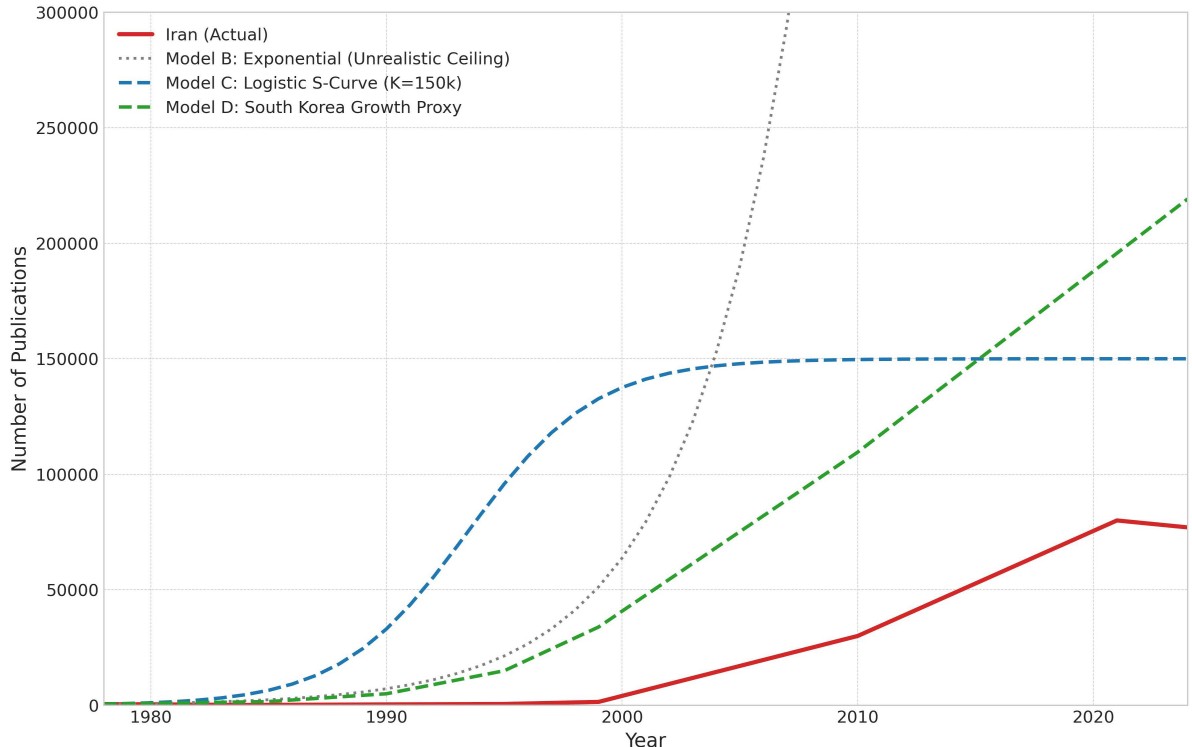

**Fig 24. Illustrative counterfactual projections for Iran's scientific output.** The solid red curve shows the actual trajectory. Model B (grey, dotted) is an exponential upper bound (unrealistic ceiling). Model C (blue, dashed) follows a logistic S-curve with carrying capacity $K = 150{,}000$. Model D (green, dashed) applies South Korea's historical year-on-year growth rates to Iran's 1978 base. The y-axis is truncated for readability; Model B continues off scale.

subsequent success represents the archetypal model of a "developmental state" that maintained political stability and executed a long-term, strategic vision for building a knowledge-based economy. Therefore, its growth trajectory serves as a credible "alternate timeline" for a nation that started from a similar baseline but did not experience a disruptive political shock.

Instead of a fixed rate, this model applies the actual, historical year-on-year growth percentage of South Korea ($g_{SK,t}$) to Iran's trajectory from 1979 onwards:

$$\hat{N}^D_{IR,t} = \hat{N}^D_{IR,t-1} \times (1 + g_{SK,t})$$

The result, as shown by the dashed green line in Fig 24, indicates that applying South Korea's historical growth dynamics to Iran's larger 1978 base yields a very high upper-range scenario, reaching an annual output of approximately 245,000 papers by 2024. This is because Iran would have been applying Korea's high growth rates to its own, larger initial base of scientific output. This data-driven simulation provides an illustrative upper-range estimate of the implied long-run shortfall under one developmental-growth proxy logic.

Collectively, the large gap between the solid red line (Iran's actual path) and the plausible scenarios modeled by the blue and green lines provides a robust, data-grounded estimate of the significant and lasting cost of the 1979 disruption.

### A.1.4 Field-specific FWCI robustness

Fig 25 provides the full year-by-year field-specific robustness view underlying the summary heatmap reported in the main text. The panel trends confirm that Iran's relative citation impact improved over time in several disciplines, but the aggregate cross-country pattern remains unequal even after field-level normalization.

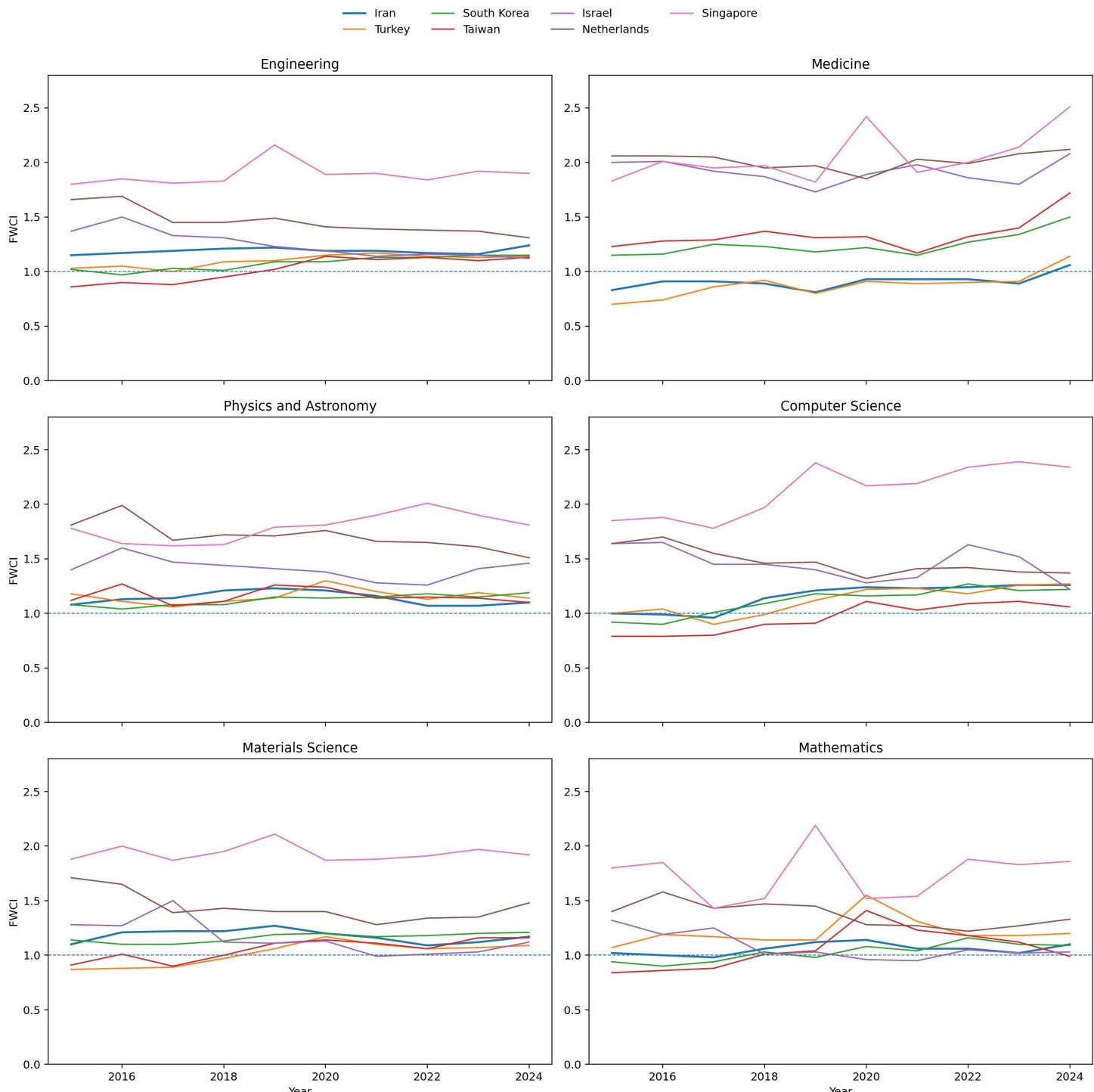

**Fig 25. Field-specific FWCI trends by country (2015–2024).** Annual FWCI trajectories across six major subject areas provide a disaggregated view of cross-country research impact. These field-level comparisons support the main interpretation in the text by showing that Iran's quality gap is reduced in some domains but does not disappear when comparisons are made within subject areas.

## Supporting information

**S1 Data. Data.**
(ZIP)

## Author contributions

**Conceptualization:** Ehsan Roohi.

**Data curation:** Ehsan Roohi.

**Formal analysis:** Ehsan Roohi.

**Investigation:** Ehsan Roohi.

**Methodology:** Ehsan Roohi.

**Resources:** Ehsan Roohi.

**Software:** Ehsan Roohi.

**Supervision:** Ehsan Roohi.

**Validation:** Ehsan Roohi.

**Visualization:** Ehsan Roohi.

**Writing – original draft:** Ehsan Roohi.

**Writing – review & editing:** Ehsan Roohi.

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
