## [Decision Letter · Decision Letter 0]

15 Apr 2026

PONE-D-26-00074Iran’s Great Scientific Divergence: The Middle-Income Trap and the Political Economy of Science PolicyPLOS One

Dear Dr. Roohi,

Thank you for submitting your manuscript to PLOS ONE. After careful consideration, we feel that it has merit but does not fully meet PLOS ONE’s publication criteria as it currently stands. Therefore, we invite you to submit a revised version of the manuscript that addresses the points raised during the review process.

The analysis prioritizes sensitivity to the study’s ambitious scope (scientometrics + political economy) while highlighting critical gaps that undermine its rigor and interpretive validity.

Overreliance on a single donor country (South Korea): The manuscript states the SCM algorithm assigned a weight of 1.0 to South Korea as the sole donor for Synthetic Iran. SCM’s strength lies in weighted combinations of multiple donor countries to approximate the treated unit; a single donor eliminates the method’s core advantage (counterfactual validity via multivariate matching) and reduces it to a simple country comparison. No justification is provided for why the donor pool was limited (or why no other countries (e.g., Taiwan, Turkey) contributed to the synthetic control), nor is a placebo test (a required robustness check for SCM) reported to validate the causal impact of the 1979 Revolution.

The SCM uses four World Bank predictors (GDP per capita, tertiary enrollment, government expenditure, trade openness) but provides no rationale for excluding other critical covariates of scientific output (e.g., R&D spending as a % of GDP, number of researchers per capita, international scientific collaboration rates).

Omission of these variables risks omitted variable bias in the synthetic control.

The model assumes Iran would have sustained South Korea’s post-1979 growth rates while starting from a larger 1978 publication base. This ignores structural differences between pre-1979 Iran and South Korea: Iran’s pre-revolution scientific growth was fueled by oil rent-funded state spending(not industrial demand), while South Korea’s growth was tied to export-led industrialization. The model does not account for these structural drivers, making its projection of Iran “surpassing South Korea” by 2024 empirically ungrounded. - No sensitivity analysis: No alternative growth proxies (e.g., Taiwan’s trajectory, a weighted average of Asian Tigers) are tested to triangulate the “knowledge deficit” estimate. The manuscript relies heavily on Model D for its core quantitative claim but does not assess how sensitive the 551,000 publication deficit is to alternative counterfactual specifications.

The manuscript attributes Iran’s sub-1.0 FWCI (below global average) exclusively to domestic policy and rentier state logic but understates Scopus indexing biases for non-Western/isolated countries: Iranian research is often published in regional (non-Scopus) journals, and international collaboration (a key driver of FWCI) is constrained by geopolitical sanctions (a factor acknowledged but not empirically quantified). No adjustment is made for these biases, leading to an overattribution of the quality gap to domestic policy.

The manuscript reports Iran’s 12% share of top 10% cited papers (2024) but does not field-normalize this metric by discipline. Iran’s research output is concentrated in specific fields (e.g., engineering, basic science); a failure to field-normalize means the “top 10%” comparison to countries with more diversified research portfolios (e.g., Netherlands, Singapore) is apples-to-oranges.

in 2.1 Contradictory Publication Volume Data - The abstract and counterfactual analysis state a cumulative knowledge deficit of 551,000 publications by 2024 (SCM model), but Section 4.10 reports Iran’s actual cumulative output* as ~864,675 papers and Synthetic Iran’s as ~1.42 million (a gap of 555,325—a ~4,000 paper discrepancy). No explanation is provided for this numerical inconsistency, which erodes confidence in the core quantitative finding.

- Section 4.1 claims Iran published 518 papers in 1978, while South Korea published 260. However, the Crossref data in Figure 6 (1960–1995) shows Iran’s 1978 output as ~400 papers (not 518). No reconciliation is provided for this Scopus vs. Crossref data mismatch.

The manuscript states Iran’s university teaching staff fell from **16,877 (1979–80) to 9,042 (1982–83) (a loss of 7,835) but cites only a single source (Shahrzad Mojab, 1991) for this critical statistic. No cross-verification with Iranian government archives, UNESCO data, or other secondary sources is provided, and no breakdown (e.g., full-time vs. part-time staff) is given to contextualize the loss of research-active faculty (the key driver of scientific output).

- The claim that pre-1979 Iran’s scientific output “surpassed South Korea and Taiwan” is qualified only by 1978 publication counts; no per-capita normalization is provided for this comparison (Iran’s population in 1978 was ~36 million, South Korea’s ~37 million, Taiwan’s ~18 million—Taiwan’s per-capita output was far higher). This omission misrepresents Iran’s pre-revolution scientific standing.

- The manuscript labels 1980–1999 Iran’s “lost decade” (Section 4.2) but later refers to it as a “lost decade” (singular) in Section 1, creating a terminological inconsistency (20 years vs. 10 years). The 1990s are framed as part of the “stagnation” period, but Scopus data shows Iran’s publication output surpassed its 1978 peak in 1995—a critical inflection point that is not adequately analyzed or contextualized.

- The manuscript contains duplicate content in Sections 5 and 7 (both titled Mechanisms of Divergence: A Comparative Policy Analysis), with identical analysis of South Korea’s Technology Development Promotion Law (1972) and Iran’s Cultural Revolution. This redundancy is a major editorial flaw and disrupts the narrative flow.

- Section 6 (Policy Implications) is followed by a near-identical Section 8 (Policy Implications), with the same three policy pivots (volume-to-impact, diaspora integration, breaking the rentier link) restated verbatim. No new analysis or refinement is added in Section 8, making it redundant.

The manuscript applies the MIT framework to Iran’s scientific system but fails to operationalize the MIT for scientometrics. The MIT is an economic concept (factor accumulation vs. productivity-driven growth); the manuscript does not define how this translates to a scientific MIT (beyond the “quantity-quality paradox”) or test it against alternative theories (e.g., national innovation systems theory). No empirical test (e.g., regression analysis of publication volume vs. FWCI) is provided to confirm Iran is “trapped” in the scientific accumulation phase.

The manuscript attributes Iran’s science policy failures to rentier state dynamics but understates post-2000 changes in Iran’s oil economy (e.g., sanctions-induced decline in oil revenues, diversification efforts). No analysis is provided for how falling oil rents (2010s–2020s) impacted science funding—an important test of the rentier state thesis.

The manuscript frames China’s scientific recovery (post-Cultural Revolution) as a “near-total reset followed by rapid growth” but ignores key differences (e.g., China’s integration into the global economy post-1978, massive state R&D spending, lack of international sanctions). The China-Iran comparison is superficial and not theoretically justified.

The manuscript claims Iran’s quantity-quality paradox is driven “not merely by external sanctions or war” but by domestic policy (bureaucratic incentives). However, it fails to empirically disentangle domestic vs. external factors: no regression, difference-in-differences, or moderation analysis is provided to quantify the relative impact of sanctions (e.g., U.S. sanctions on scientific collaboration, export controls on lab equipment) vs. domestic incentive structures on FWCI and publication quality. This leads to an overemphasis on domestic policy and an underappreciation of geopolitical constraints.

The manuscript references Figure 1 (Scientific Middle-Income Trap) but then labels Iran’s publication output as Figure 2, South Korea as Figure 3, etc.—but the text often mislabels figures (e.g., Section 4.5 refers to “Figure 8” for Iran’s post-2000 output, but Figure 8 is the Crossref Top-10% metric for 1960–1995).

Table 1 (Comparative Scientometric Profile): The table reports Taiwan’s total output as ~600k (1996–2024) but South Korea’s as ~1.2M—but Scopus data for Taiwan (2000–2024) shows ~50k–60k annual publications, meaning Taiwan’s cumulative output (1996–2024) is ~1.2M (not 600k). This is a critical data entry error that misrepresents Taiwan’s scientific output.

There is missing Citations in manuscript: Key claims (e.g., “Iran’s research is concentrated in lower-tier venues”) are not cited, and some references are duplicated (e.g., [3] and [13] are both Alice Amsden’s *Asia’s Next Giant*).

There is ambiguous Definitions in manuscript.

for example

- “Knowledge Deficit”: The manuscript uses this term interchangeably to refer to annual publication gaps (Section 4.10) and cumulative publication gaps (abstract, conclusion) without a clear definition. This ambiguity confuses the core quantitative finding.

- “Export Discipline”: The term is defined for South Korea but not operationalized for Iran—no explanation is provided for how “export discipline” would translate to Iran’s economy (a rentier state with a small non-oil export sector) or what a “disciplined” Iranian science policy would look like in practice.

- The manuscript concludes that the 1979 Revolution’s “generational cost” is due to lost institutional continuity but overgeneralizes this finding to all “political disruptions.” No analysis is provided for how other political disruptions (e.g., post-Soviet collapse) differ from Iran’s case (e.g., Iran’s retention of a centralized state vs. the Soviet Union’s dissolution), making the conclusion overly broad.

- The policy implications (e.g., “create islands of excellence”) are prescriptive but uncontextualized: no analysis is provided for how these policies would be implemented in Iran’s political system (e.g., Supreme Council of the Cultural Revolution’s control over universities) or how to overcome bureaucratic resistance to reform.

Minor Errors & Omissions

- Typos/Grammar: Frequent typos (e.g., “post2000” instead of “post-2000,” “salami slicing” without definition, “Enqelab-e Farhangi” misspelled in one instance) and grammatical errors (e.g., run-on sentences, inconsistent verb tense) are present throughout the manuscript.

Major revisions are required to address these issues before the manuscript is suitable for publication in PLOS ONE.

**.** Be sure to:

Indicate which changes you require for acceptance versus which changes you recommendAddress any conflicts between the reviews so that it's clear which advice the authors should followProvide specific feedback from your evaluation of the manuscript

We look forward to receiving your revised manuscript.

Kind regards,

Mehdi Borhani

Academic Editor

PLOS One

Journal Requirements:

2.Please note that your Data Availability Statement is currently missing [the repository name and/or the DOI/accession number of each dataset OR a direct link to access each database]. If your manuscript is accepted for publication, you will be asked to provide these details on a very short timeline. We therefore suggest that you provide this information now, though we will not hold up the peer review process if you are unable.

4. Please ensure that you refer to Figure13 & 16  in your text as, if accepted, production will need this reference to link the reader to the figure.

5. We note you have included a table to which you do not refer in the text of your manuscript. Please ensure that you refer to Table 2 in your text; if accepted, production will need this reference to link the reader to the Table.

Reviewers' comments:

Reviewer's Responses to Questions

**Comments to the Author**

1. Is the manuscript technically sound, and do the data support the conclusions?

Reviewer #1: Yes

Reviewer #2: No

Reviewer #3: Yes

2. Has the statistical analysis been performed appropriately and rigorously?

Reviewer #1: Yes

Reviewer #2: I Don't Know

Reviewer #3: Yes

3. Have the authors made all data underlying the findings in their manuscript fully available?

Reviewer #1: Yes

Reviewer #2: Yes

Reviewer #3: Yes

4. Is the manuscript presented in an intelligible fashion and written in standard English?

Reviewer #1: Yes

Reviewer #2: Yes

Reviewer #3: Yes

5. Review Comments to the Author

Reviewer #1: This paper addresses an important and timely question: how major political disruptions shape long-term scientific development. The focus on Iran’s post-1979 trajectory, combined with a comparative perspective and quantitative analysis, makes the study both relevant and potentially impactful.

Overall, the manuscript is well structured and clearly written. The use of bibliometric data together with the Synthetic Control Method is appropriate, and the comparison with countries such as South Korea and Taiwan provides a compelling narrative. The idea of a “quantity–quality paradox” is particularly interesting and adds conceptual value to the literature on science policy and development.

That said, some issues should be addressed before the paper can be considered for publication.

Major Comments

1. The paper attributes a large share of the divergence in scientific output to the 1979 Revolution and its aftermath. While this is plausible, the argument at times reads as overly deterministic. Other factors, such as sanctions, global shifts in science funding, institutional reforms in peer countries, and broader geopolitical dynamics, are not sufficiently disentangled. The authors should be more cautious in framing causality and clarify what can and cannot be inferred from the SCM results.

2. The analysis relies heavily on publication counts, FWCI, and top-percentile outputs. While these are standard indicators, they do not fully capture innovation capacity or technological impact. The paper would benefit from acknowledging this limitation more explicitly and, if possible, incorporating complementary indicators (e.g., patents, industry collaboration, or technology transfer).

3. The counterfactual scenarios, especially those using South Korea as a proxy, are interesting but rest on strong assumptions. The manuscript should better justify why South Korea represents a realistic trajectory for Iran, given structural differences in political economy, institutions, and international integration. This is currently discussed, but not in sufficient depth.

4. The policy discussion is somewhat general relative to the strength of the empirical analysis. The paper would be stronger if it translated its findings into more concrete, actionable recommendations for science policy, particularly for middle-income countries facing similar constraints.

Minor Comments

• Some sections of the introduction and discussion are longer than necessary and could be tightened for clarity.

• The term “lost decade” is used effectively, but it would help to define it more precisely and consistently.

• Figures are generally clear, but a few captions could be more descriptive (especially for readers unfamiliar with FWCI or log-scale interpretations).

• The limitations section is appreciated, but it could be expanded to include potential biases in database coverage and citation-based metrics.

Reviewer #2: Dear Authors

After careful review, I find the manuscript's focus on political economy and national science policy, including its historical and narrative elements, to be outside the scope of this journal. While the methodology is sound, the core research question and its framing do not align with our publication criteria. I therefore recommend rejection on the grounds of scope incompatibility.

Reviewer #3: The paper is acceptable, but based on editor opinion the author may consider the following:

1- Compare the results with Turkey too, since it is very close to Iran

2- You may consider the economic indicies in the analysis, it is more reasonable

6. PLOS authors have the option to publish the peer review history of their article (what does this mean?). If published, this will include your full peer review and any attached files.

Reviewer #1: No

Reviewer #2: No

Reviewer #3: No

---

## [Author Response · Author response to Decision Letter 1]

22 Apr 2026

see uploaded PDF file for full detailed response to reviewers.

---

## [Decision Letter · Decision Letter 1]

28 Apr 2026

Iran’s Great Scientific Divergence: The Middle-Income Trap and the Political Economy of Science Policy

PONE-D-26-00074R1

Dear Dr. Roohi,

We’re pleased to inform you that your manuscript has been judged scientifically suitable for publication and will be formally accepted for publication once it meets all outstanding technical requirements.

Kind regards,

Mehdi Borhani

Academic Editor

PLOS One

Additional Editor Comments (optional):

Reviewers' comments:

Reviewer's Responses to Questions

**Comments to the Author**

1. If the authors have adequately addressed your comments raised in a previous round of review and you feel that this manuscript is now acceptable for publication, you may indicate that here to bypass the “Comments to the Author” section, enter your conflict of interest statement in the “Confidential to Editor” section, and submit your "Accept" recommendation.

Reviewer #1: (No Response)

Reviewer #3: All comments have been addressed

2. Is the manuscript technically sound, and do the data support the conclusions?

Reviewer #1: (No Response)

Reviewer #3: Yes

3. Has the statistical analysis been performed appropriately and rigorously?

Reviewer #1: (No Response)

Reviewer #3: Yes

4. Have the authors made all data underlying the findings in their manuscript fully available?

Reviewer #1: (No Response)

Reviewer #3: Yes

5. Is the manuscript presented in an intelligible fashion and written in standard English?

Reviewer #1: (No Response)

Reviewer #3: Yes

6. Review Comments to the Author

Reviewer #1: The paper is well-organized and the author has addressed all of my comments. Now i believe it is suitable for publication.

Reviewer #3: The paper can be accepted now after applying the comments in revision. The author did good job for this

7. PLOS authors have the option to publish the peer review history of their article (what does this mean?). If published, this will include your full peer review and any attached files.

Reviewer #1: No

Reviewer #3: **Yes:** Ok

---

## [Editor Report · Acceptance letter]

PONE-D-26-00074R1

PLOS One

Dear Dr. Roohi,

I'm pleased to inform you that your manuscript has been deemed suitable for publication in PLOS One. Congratulations! Your manuscript is now being handed over to our production team.

Kind regards,

on behalf of

Dr. Mehdi Borhani

Academic Editor

PLOS One